# ROYAL SOCIETY
# OPEN SCIENCE

palaeontology

Balaenidae, new genus and species, Zanclean, *Balaenula balaenopsis*, supramastoid crest

**Author for correspondence:**
Yoshihiro Tanaka
e-mail: tanaka@mus-nh.city.osaka.jp

# A new member of fossil balaenid (Mysticeti, Cetacea) from the early Pliocene of Hokkaido, Japan

Yoshihiro Tanaka[1,2,3], Hitoshi Furusawa[4]
and Masaichi Kimura[3,5]

[1]Osaka Museum of Natural History, Nagai Park 1-23, Higashi-Sumiyoshi-ku, Osaka 546-0034, Japan
[2]Division of Academic Resources and Specimens, Hokkaido University Museum, Sapporo, Hokkaido, Japan
[3]Numata Fossil Museum, Numata, Hokkaido, Japan
[4]Sapporo Museum Activity Center, Sapporo, Hokkaido, Japan
[5]Hokkaido University of Education, Sapporo, Hokkaido, Japan

YT, 0000-0001-9598-5583

The family Balaenidae includes two genus and four extant species. Extinct balaenids are known for at least four genus and 10 species. The oldest known record of members of the Balaenidae is known from the early Miocene, but still need more early members of the family to provide better phylogenetic hypotheses. FCCP 1049 from the lower part of the Chippubetsu Formation, Fukagawa Group (3.5–5.2 Ma, Zanclean, early Pliocene) was preliminary described and identified as *Balaenula* sp. by Furusawa and Kimura in 1982. Later works discussed that FCCP 1049 is different from the genus, and is placed in different clade from *Balaenula astensis*. The result of our phylogenetic analysis places FCCP 1049 basal to *Balaenella brachyrhynus*, and is again separated from *B. astensis*. In this study, FCCP 1049 is re-described and named as *Archaeobalaena dosanko* gen. et sp. nov. *Archaeobalaena dosanko* is distinguishable from other balaenids by having a deep promontorial groove of the pars cochlearis of the periotic. *Archaeobalaena dosanko* can be differentiated from other balaenids, except *Morenocetus parvus* by having a slender zygomatic process, and posteriorly oriented postorbital process in dorsal view. *Archaeobalaena dosanko* adds detailed skull, periotic and bulla morphologies for the earlier balaenids.

## 1. Introduction

The family Balaenidae includes two genus and four extant species. Extinct balaenids are known for at least four genus and 10 species [1–3]. The modern balaenid body length reaches 17–20 m [4]. The

oldest known nominal species of the members of the Balaenidae is known from the early Miocene of Patagonia, Argentina [1], and its body length was estimated as 4.8–6.2 m [5]. Our knowledge of balaenid past diversity is growing [1,3,5–8], but still needs earlier members of the family to hypothesize their phylogeny.

A Pliocene balaenid, the genus *Balaenula* had been treated as 'a taxonomical basket, where all the small-sized balaenids were put' in history [2]. The type species of the genus *Balaenula balaenopsis* from Antwerp was established by Van Beneden [9], but the holotype is doubtfully recognized as an individual [1]. Supposed similar cases of establishing new species with mixed individuals was centered on a cetotheriid, *Herpetocetus scaldiensis* of Van Beneden [9], which was summarized by Deméré *et al.* [10]. The second species, *Balaenula astensis* from the late early Pliocene of Villafranca d'Asti, was established by Trevisan [11] and re-described by Bisconti [8].

Furusawa & Kimura [12] preliminary identified FCCP 1149 from an early Pliocene sediment in Hokkaido, Japan, as *Balaenula* sp. based on having a low triangle-shaped occipital with a depression at the centre, an acute angle of the nuchal crest against the plain in posterior view and a rounded exoccipital placing below to the ventral margin of the occipital condyle. The study mentioned that these features can be seen only on *B. balaenopsis*. Later, Bisconti [8] mentioned that FCCP 1049 is different from *Balaenula* based on an extended temporal fossa (p. 47). Indeed, several diagnoses of the genus *Balaenula* were reported by Bisconti [2]. One of the diagnoses for the genus, not having a protruding nuchal crest is not seen on FCCP 1049. FCCP 1049 shows a lateral expansion at the posterior part of the supraoccipital crest. One of the most recent phylogeny works found that FCCP 1049 did not form a clade with *B. astensis* [1]. To expand diversity and morphological information for understanding the earlier balaenid evolution, we update the identification of FCCP 1049 and its geological age, and also re-describe the specimen.

## 2. Material and methods

Morphological terminology follows mainly Mead & Fordyce [13] and Ekdale *et al.* [14] for some earbone terms.

### 2.1. Institutional abbreviations

FCCP, Fukagawa City Cultural Properties, Hokkaido, Japan. HUES, Hokkaido University of Education Sapporo Campus, Hokkaido, Japan. IMNH, Icelandic Museum of Natural History. IRSNB, Institut Royal des Sciences Naturelles de Belgique, Brussels, Belgium. MLP, Museo de La Plata, La Plata, Argentina. USNM, National Museum of Natural History, Smithsonian Institution, Washington, DC.

## 3. Systematic palaeontology

Cetacea [15]
Neoceti [16]
Mysticeti [17]
Chaeomysticeti [18]
Balaenidae [19]
*Archaeobalaena* gen. nov
*LSID*. urn:lsid:zoobank.org:act:BA112B5B-7FCB-4882-B689-1BB238F4F5CC.
*Typespecies. Archaeobalaena dosanko* sp. nov
*Etymology*. The generic name, *Archaeobalaena*, is named in Greek archaios meaning ancient, and the type genus name of the family Balaenidae.
*Diagnosis.* As for the holotype and only species.
*Archaeobalaena dosanko* sp. nov
*LSID*. urn:lsid:zoobank.org:act:0FB35AF7-ED4A-4987-8EA5-A9E290D89A6C.
*Holotype.* FCCP 1049, including the maxilla, frontal, parietal, squamosal, exoccipital, supraoccipital, periotics, tympanic bullae, right mandible, two thoracic vertebrae, a caudal vertebra, presternum and five ribs, but no rostrum. FCCP 1049 was previously registered as HUES 100003 [12].
*Locality and horizon.* FCCP 1049 was found at a river bed of Tadoshi River in Fukagawa City, Hokkaido, Japan, by J. Takahashi, N. Mita and others on 22 September 1978 and dug up in 1979 and 1980 by over 100 people [20]: latitude 43°47′36.31″ N, longitude 142°5′10.00″ E (figure 1).

FCCP 1049 was found in the lower part of the Chippubetsu Formation, Fukagawa Group, especially lower to so-called T1 tuff layer [23]. Regarding the study, there is a T2 tuff layer above the T1 tuff layer. The T2 tuff layer is correlated to S1 tuff of the Takikawa Formation. The age of S1 tuff is 4.1 ± 0.6 Ma [24]. T1 tuff possibly distributes above Ops tuff of the upper part of the Horokaoshirarika Formation

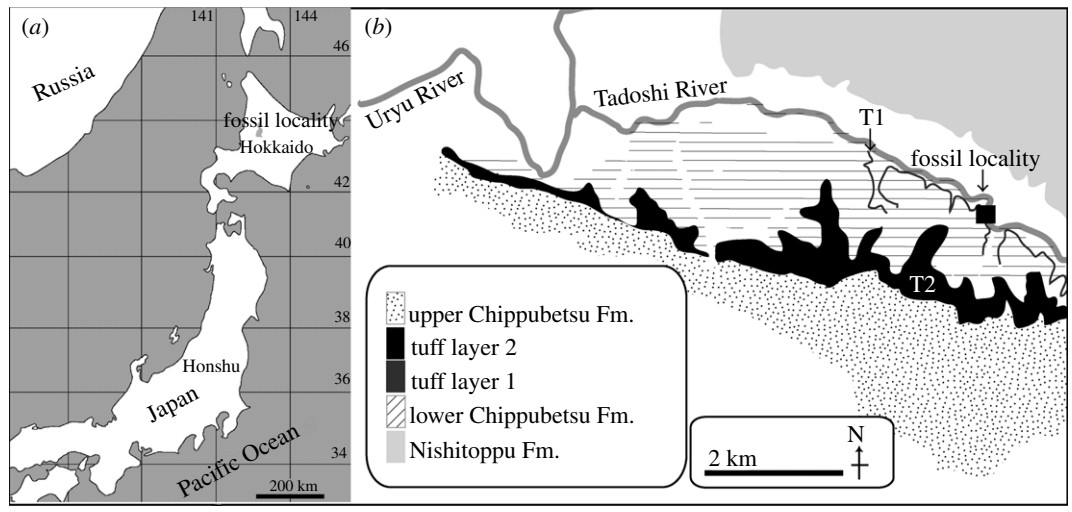

**Figure 1.** Maps showing the locality of FCCP 1049, *A. dosanko*. The base maps (*a*) and (*b*) are modified from Tanaka & Kohno [21] and Wada & Akiyama [22].

[23]. The age of Ops tuff is 4.5 ± 0.7 Ma. Thus, the age of FCCP 1049 can be taken as 3.5–5.2 Ma (Zanclean, early Pliocene) with wider ranges. Very near to the locality (about 10–20 km away), *Numataphocoena yamashitai* and Herpetocetinae gen. et sp. indet. and some Mysticeti indet. materials from the early Pliocene (about 4.5–3.5 Ma) possibly be cetaceans of the same age [25–29]. From the same area, there are some late Miocene Mysticeti reports: type specimen of *Miobalaenoptera numataensis* (6.5–6.8 Ma) and a referred specimen *Herpetocetus* sp. (7.7–6.8 Ma), but are not simultaneous records of FCCP 1049 [30,31].

*Etymology.* Dosanko means people and things born in Hokkaido, northern Japan, originating from the native horse of Hokkaido.

*Diagnosis.* Among the Balaenidae, *A. dosanko* uniquely has a deep promontorial groove of the pars cochlearis of the periotic (character 151). *Archaeobalaena dosanko* can be differentiated from other balaenids, except *Morenocetus parvus* by having a slender zygomatic process, and posteriorly oriented postorbital process in dorsal view (character 38). *Archaeobalaena dosanko* can be differentiated from *M. parvus* by having a slender and laterally tilted down of the supraorbital process. *Archaeobalaena dosanko* can be differentiated from all balaenids, except the *Balaena* and *Eubalaena* by having a weakly and laterally projected lateral margin of the nuchal crest.

Comparison with more basal balaenids (*M. parvus* and *Peripolocetus vexillifer*), *A. dosanko* can be differentiated by having a more or less same length of the anterior process and pars cochlearis of the periotic (character 139), hypertrophied and blade-like lateral tuberosity (character 144), posterior process of the periotic orienting a right angle to the axis of the anterior process in ventral view (character 170), laterally reduced involucral ridge in dorsal view (character 181), the dorsolateral surface of involucrum forming a continuous rim (character 190), and a flat anteromedial portion of the ventral surface of tympanic bulla (character 195). Comparison with more late diverging balaenids (*B. astensis*, *Balaenella brachyrhynus* and so on), *A. dosanko* can be differentiated by having the medial lobe of the tympanic bulla (character 187). Comparison with crown balaenids (*Balaena* spp. and *Eubalaena* spp.), *A. dosanko* can be differentiated by having a thickened and flat lateral surface of the orbital rim of the supraorbital process (character 40), pyramidal process (character 141), laterally exposed and distinct compound posterior process from the lateral skull wall (character 172), no crest on the parieto-squamosal suture (character 93), no hypertrophied suprameatal fossa (character 162), and no transverse creases on the dorsal surface of the involucrum (character 191).

# 4. Description

## 4.1. Ontogeny

FCCP 1049 possibly is subadult, because it shows a caudal vertebra with fused epiphyses, thoracic vertebrae with unfused epiphyses and isolated vertebral epiphyses [32]. On the skull, the exoccipital/supraoccipital suture is fused, which suggests that FCCP 1049 is older than postnatal [33]. The frontal/parietal, parietal/squamosal, squamosal/exoccipital sutures are fused but visible.

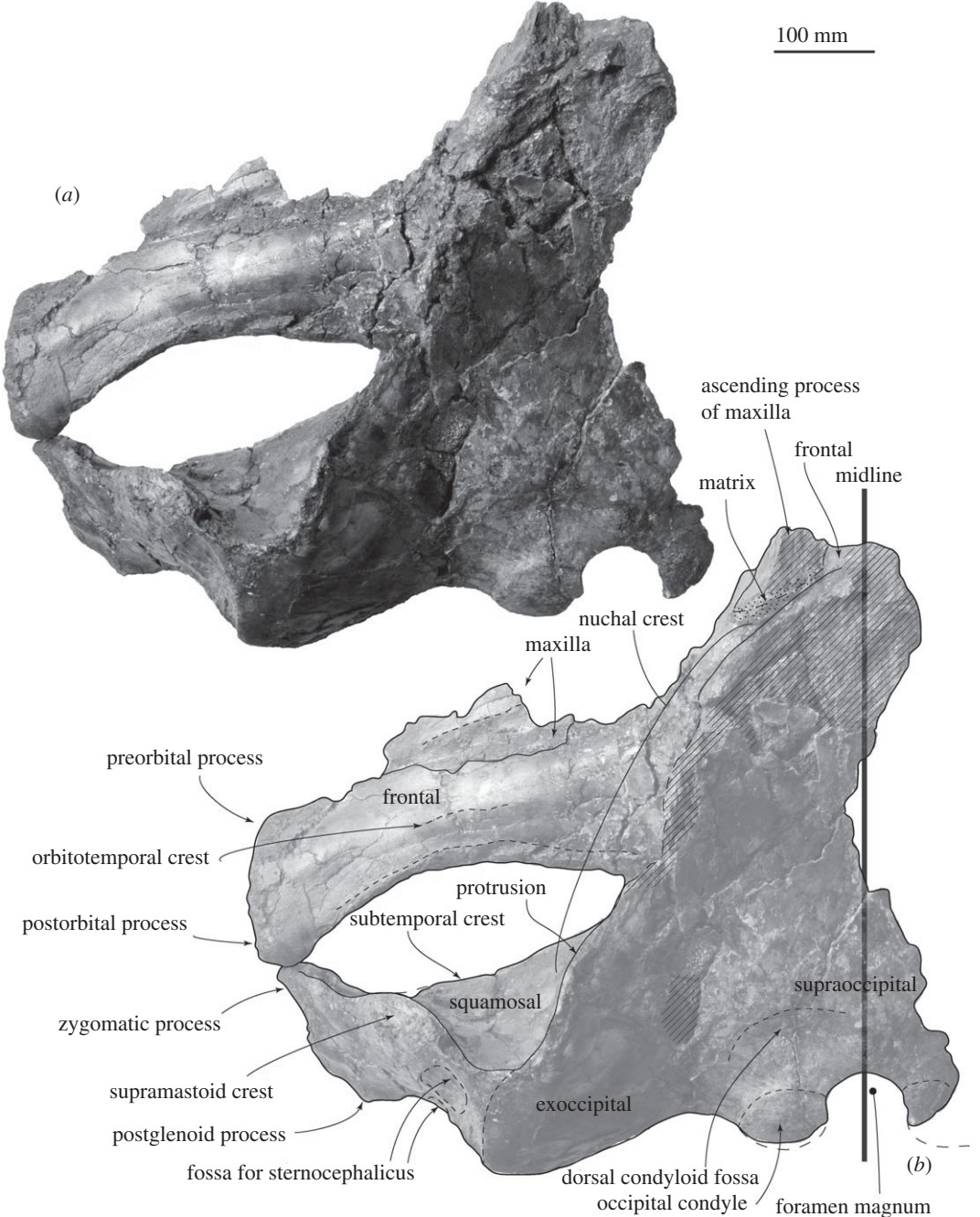

**Figure 2.** Skull of FCCP 1049, *A. dosanko* in dorsal view. (*a*) Photo and (*b*) line art.

## 4.2. Skull

*General features of the skull.* A preserved left side of the skull has a slender supraorbital process with weakly projected pre and postorbital processes, slender zygomatic process and laterally weakly protruded nuchal crest. The temporal fossa is anteroposteriorly short and mediolaterally wide.

*Maxilla.* A supposed ascending process is wide and flat (about 87 mm wide, figures 2 and 3). Its posterior end still has a matrix and does not show the posterior border. The infraorbital plate covers the anterior border of the supraorbital process. The plate has a transverse ridge on the dorsal surface.

*Frontal.* The supraorbital process is slender and tilts down laterally. The mid part of the supraorbital process is anteroposteriorly shorter, and the lateral part is anteroposteriorly longer, because the posterior border of the supraorbital process is anteriorly excavated at the middle (figure 2). There is a weak transverse orbitotemporal crest on the dorsal surface of the supraorbital process. The lateral margin of the orbit in dorsal view is more or less straight. In lateral view (figure 4), an

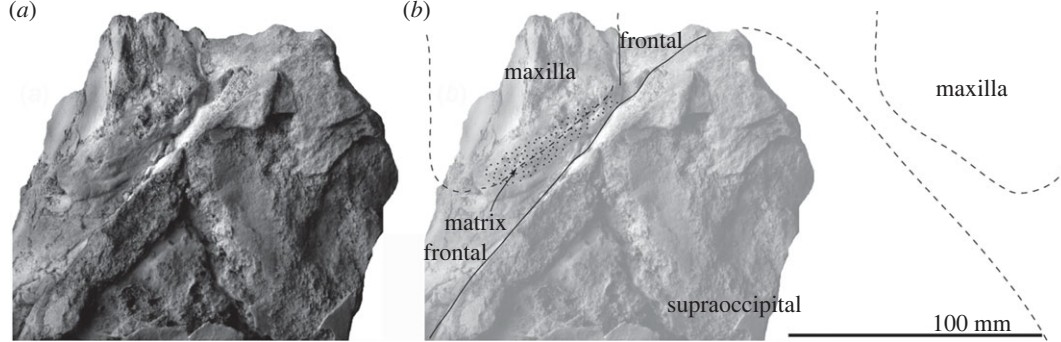

**Figure 3.** Skull, vertex of FCCP 1049, *A. dosanko* in dorsal view. (*a*) Photo and (*b*) line art.

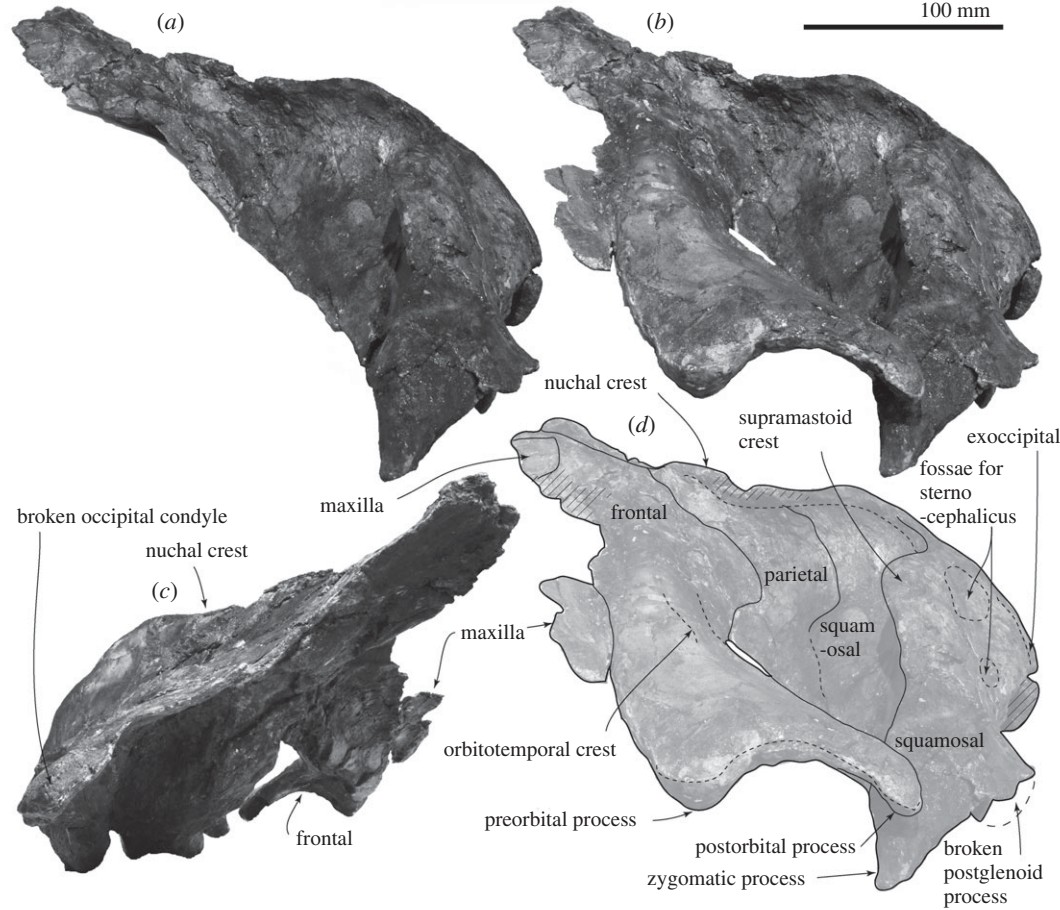

**Figure 4.** Skull of FCCP 1049, *A. dosanko* in lateral view. (*a*) Photo without the supraorbital process in left lateral view; (*b*) photo with the supraorbital process in left lateral view; (*c*) photo in right lateral view; (*d*) line art of (*b*).

anteroposteriorly thin postorbital process projects posteroventrally, and it has a rounded outline in lateral view. The preorbital process has a rounded outline and is more robust than the postorbital process. Between the preorbital and postorbital processes, a dorsally excavated orbital rim is formed. The medial end of the frontal contacts with the parietal posteriorly at the anterior portion of the temporal fossa. The frontal forms a part of the vertex, and is partially covered by the ascending process of the maxilla. Medial to the ascending process, the frontal is exposed at a supposed centre of the vertex, but the interfrontal suture is not preserved. The frontal ridge might be absent, not like *M. parvus* [1]. In ventral view, the optic canal has a short and deep medial part and long and shallow lateral part, and is restricted by the pre and postorbital ridges running mediolaterally.

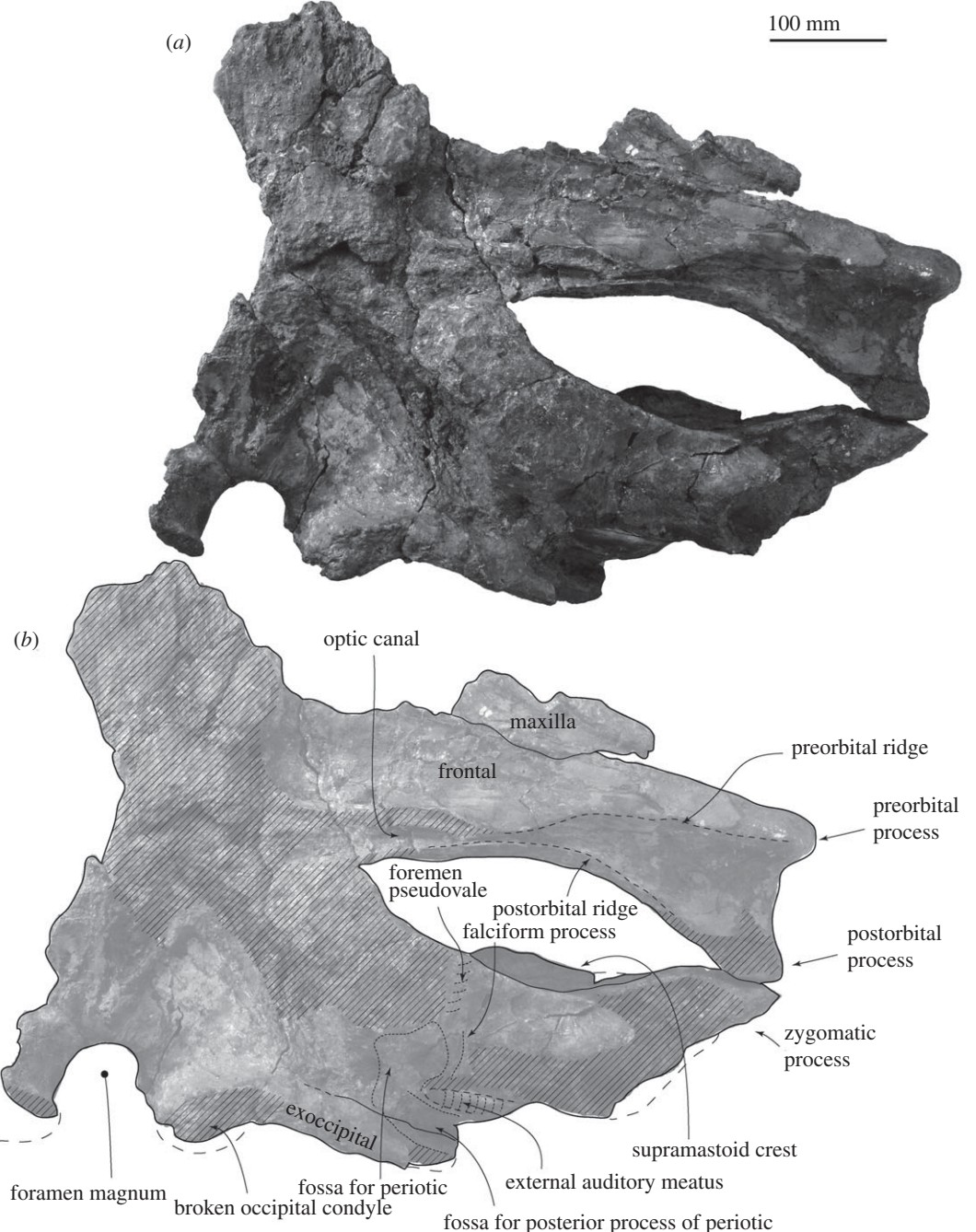

**Figure 5.** Skull of FCCP 1049, *A. dosanko* in ventral view. (*a*) Photo and (*b*) line art.

*Parietal.* The parietal locates posterior to the frontal and the suture shows an angle in lateral view. The parietal forms the anteromedial wall of the temporal fossa (figure 4), which is weakly concave. The parietal/squamosal suture is visible.

*Squamosal.* In dorsal view, the squamosal has a slender zygomatic process projecting anterolaterally (figure 2). The anterior part of the zygomatic process is anteroposteriorly thin, and forms a flat anterior surface for a contact with the postorbital process. On the dorsal surface of the zygomatic process, a strong supramastoid crest runs posteromedially and reaches the posterior part of the nuchal crest. The supramastoid crest has a huge anteroposteriorly thin rounded squamosal prominence, dorsal to the postglenoid process and external auditory meatus. The mediolateral length of the crest is about 170 mm and dorsoventral height is about 65 mm. The posterodorsal surface of the squamosal shows two square shallow fossae for the sternocephalicus.

In ventral view (figure 5), the squamosal has a very faint falciform process, which could be identified using the periotic (figure 6). Anterior to the falciform process, there is an anteroposteriorly long shallow

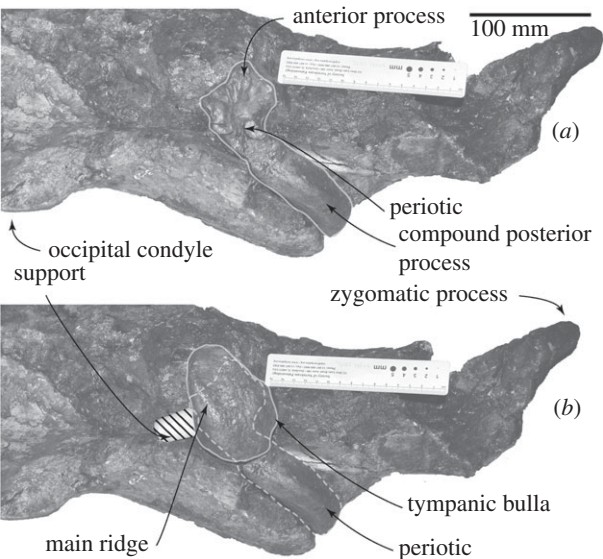

**Figure 6.** Skull and left ear bones of FCCP 1049, *A. dosanko* in ventral view. (*a*) With periotic and without tympanic bulla and (*b*) with periotic and tympanic bulla *in situ*.

**Figure 7.** Skull of FCCP 1049, *A. dosanko* in anterior and posterior views. (*a*) Photo; (*b*) line art; (*c*) photo; (*d*) line art.

groove, which might be a part of the foramen pseudovale. A shallow fossa for the periotic locates medial to the falciform process.

Posterior to the falciform process and lateral to the fossa for the periotic, a large external auditory meatus (23.5 mm long, 26.0+ mm deep at the lateral end of the meatus) runs transversely. Posterior to

anterior process of periotic

sigmoid process

50 mm

posterior pedicle

compound posterior process

**Figure 8.** Right periotic and tympanic bulla connections of FCCP 1049, *A. dosanko* in ventrolateral view.

the external auditory meatus, there is a huge laterally wider fossa for the compound posterior process of the periotic (29.0 mm long, 32.0 mm high at the lateral end of the fossa). The fossa shows the border between the squamosal and exoccipital. The base of the postglenoid process is wide even though its anterior and ventral parts are broken away in posterior view (figure 7*a*,*b*).

*Exoccipital and supraoccipital.* In dorsal and posterior views (figures 2 and 7*a*,*b*), the exoccipital is wide and occupies about 66% of the bizygomatic width. The exoccipital forms a rounded ventrolateral part of the occipital shield. In ventral view, the exoccipital is a mediolaterally long plate (about 27 mm long), but the medial end is strongly worn. The lateral part of the exoccipital forms a posterior part of the fossa for the posterior process. The occipital condyle is flat and only weakly projects posteriorly. There is a shallow dorsal condyloid fossa dorsal to the occipital condyle. The foramen magnum preserves dorsal side and its dorsal margin becomes thicker. The supraoccipital is a long triangle (figure 2). The lateral margin of the supraoccipital and exoccipital is incomplete, but shows a small protrusion slightly dorsal to the level of the supramastoid crest

## 4.3. Periotics

The periotics (figures 6 and 8–10 and table 1) have a robust anterior process, small globular pars cochlearis and large compound posterior process of the tympanoperiotic.

The anterior process is short. The size and shape of the lateral tuberosities and also the posterior processes are slightly different on the right and left sides (table 1). The right one is a broad triangle with an extension that projects laterally. The left one is smaller than the right side. Anterior to the lateral tuberosity, there is a shallow notch, which might be the anteroexternal sulcus. Medial to the lateral tuberosity, the anterior pedicle of the tympanic bulla is anteroposteriorly long, rectangular, and is placed on the ventral surface of the anterior process. Between the anterior pedicle and lateral tuberosity, there is a large and weakly depressed plane for the sigmoid process of the tympanic bulla (figure 8). A shallow wide mallear fossa is located slightly posterior to the level of the lateral tuberosity. In dorsal view, the medial part of the periotic has two large processes: the pyramidal process anteriorly and the dorsal tuberosity posteriorly. Between these processes, is the anteroposteriorly long and strongly curved dorsal rim of the suprameatal fossa [35]. Ventral to the rim, the suprameatal fossa is large and shallow.

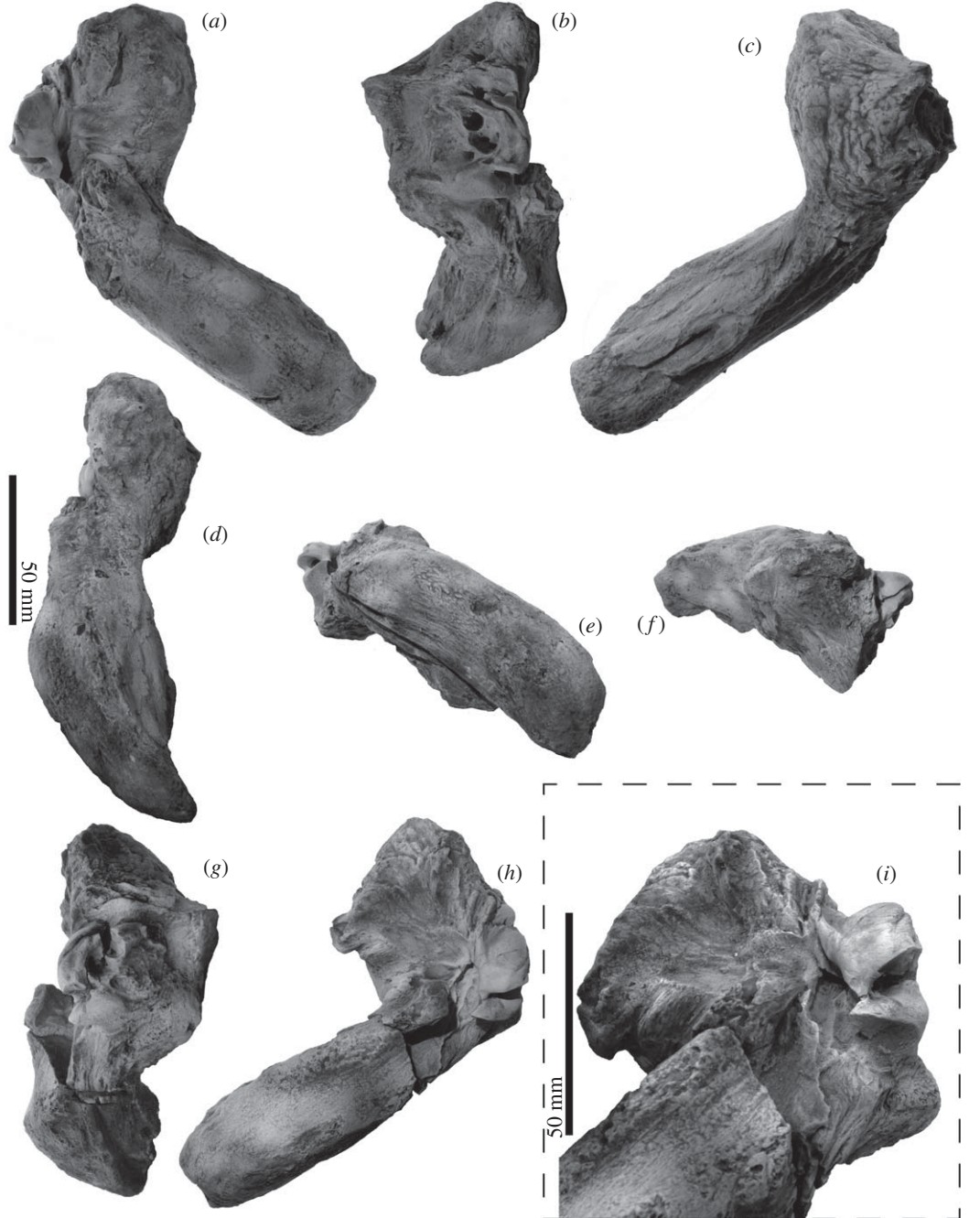

**Figure 9.** Periotics of FCCP 1049, *A. dosanko*; (*a–f*) left periotic; (*g–i*) right periotic; (*a,h*) ventral view; (*b,g*) medial view; (*c*) dorsal view; (*d*) lateral view; (*e*) posterior view; (*f*) anterior view; (*i*) posteroventral view.

An anteroposteriorly long globular pars cochlearis covers the body of the periotic anteriorly as a thin bone and it forms the anterior incisure. In the anterior incisure, there is a huge dorsoventrally long elliptical hiatus fallopii (4.7 mm high, 3.2 mm wide) opening anteriorly. The medial surface of the pars cochlearis has a deep, anteroposteriorly long and weakly curved promontorial groove. Just posterior to the hiatus fallopii, a large internal acoustic meatus (9.1 mm long, 5.2 mm high) opens medially and contains the proximal opening of the facial canal and dorsal vestibular area. The proximal opening of the facial canal is more or less circular (5.0 mm high, 4.3 mm long). Just posterior to the facial canal, and clearly separated from it by a low transverse crest, there is a circular dorsal vestibular area (5.2 mm high, 5.9 mm long). In the dorsal vestibular area, there is a small foramen singulare (1.0 mm in diameter) anterodorsally and two depressed areas, which might be the spiral cribriform tract and area cribrosa media. The spiral cribriform tract and area cribrosa media are connected and form a weakly curved depression. Posterior to these openings, there are two foramina:

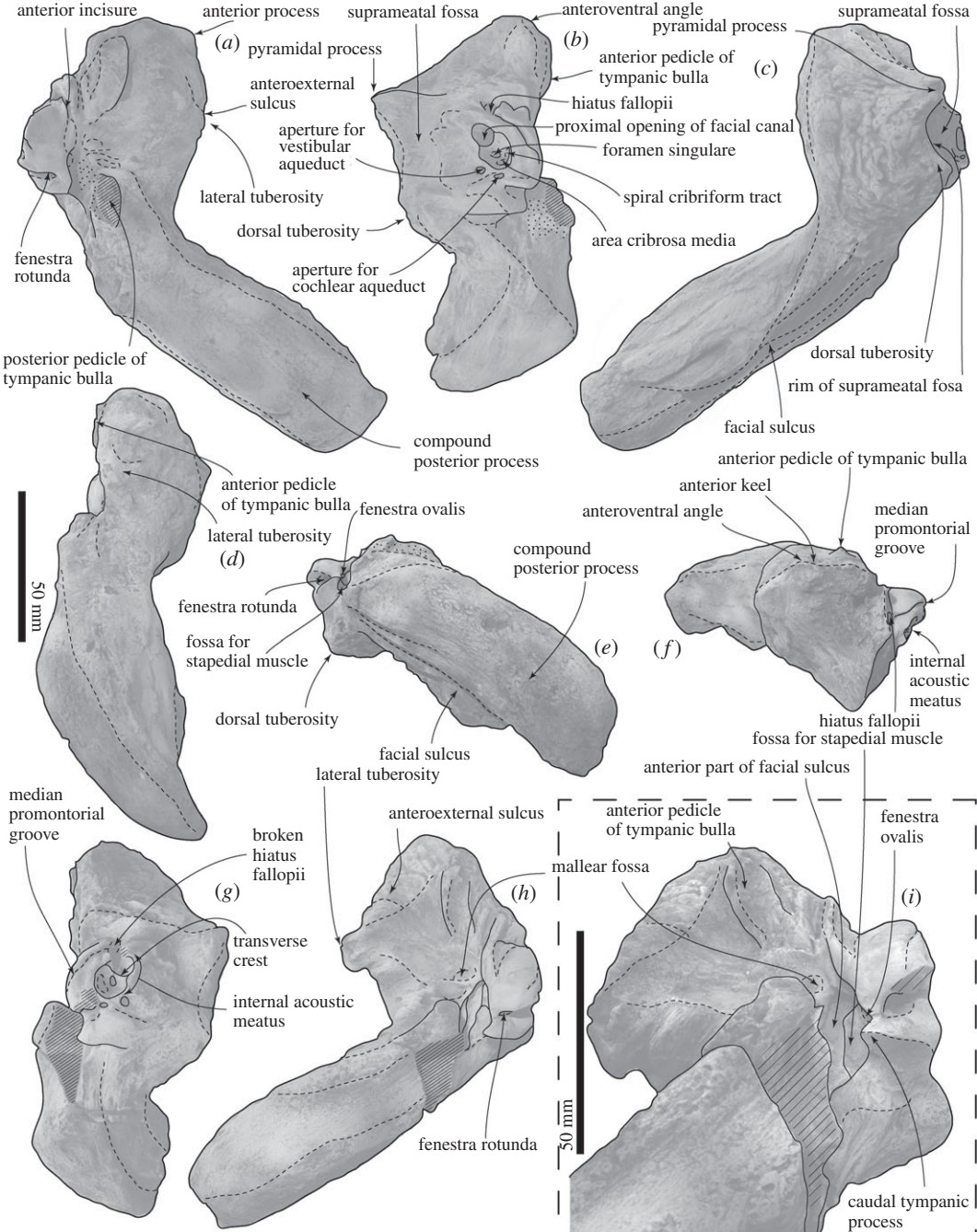

**Figure 10.** Key features of periotics of FCCP 1049, *A. dosanko*; (*a*–*f*) left periotic; (*g*–*i*) right periotic; (*a*) and (*h*) ventral view; (*b*,*g*) medial view; (*c*) dorsal view; (*d*) lateral view; (*e*) posterior view; (*f*) anterior view; (*i*) posteroventral view.

a larger aperture for the vestibular aqueduct (4.0 mm high, 4.8 mm long) and a smaller aperture for the cochlear aqueduct (2.5 mm high, 1.2 mm long). The posterior end of the pars cochlearis has a large elliptical fenestra rotunda (4.4 mm long, 8.1 mm wide), and the anterior border of the fenestra rotunda runs to the medial end of the pars cochlearis and projects as an anteroposteriorly short plate. The posterior end of the pars cochlearis is formed by the caudal tympanic process, which has transversely long sharp edge.

A large fenestra ovalis (6.0 mm long, 4.3 mm wide) is located posterior to the mallear fossa, lateral to the pars cochlearis (figure 10*i*). Posterior to the fenestra ovalis, there is a deep and long stapedial muscle fossa (15.5 mm long, 4.5 mm wide). Medial to the fenestra ovalis, there is a long fossa for the anterior part of the facial sulcus. The facial sulcus runs posteriorly, and forms a deep groove on the posterior surface of the compound posterior process of the tympanoperiotic. Lateral to the facial sulcus, an anteroposteriorly

**Table 1.** Measurements in millimetres of *A. dosanko* (FCCP 1049) periotic and tympanic bulla. (Dimensions follow [34]. Distances are either horizontal or vertical, unless identified as point to point. Measurements of *Balaenula balaenopsis* IRSNB CtM 858a were taken from table 7 of Bisconti [2].)

| periotic | left | right | *Balaenula balaenopsis* IRSNB Ct. M. 858a |
|---|---|---|---|
| length of anterior process of periotic | 34.7 | 36.0 | 32.0 |
| width of anterior process of periotic | 46.4 | 60.0 | 43.5 |
| length of anterior process, from anteroventral angle to anterior incisure | 24.3 | 32.7 | — |
| maximum transverse width of the body of periotic | 46.4 | 60.0 | — |
| length of pars cochlearis | 30.2 | 31.4 | 54.0 |
| width of pars cochlearis | 16.8 | 17.7 | 30.0 |
| dorsoventral diameter of internal acoustic meatus | 9.3 | 9.3 | 4.3 |
| length of internal acoustic meatus | 13.7 | 13.9 | 6.8 |
| greatest length of aperture for the cochlear aqueduct | 1.2 | — | — |
| greatest length of aperture for the vestibular aqueduct | 4.8 | 5.0 | — |
| dorsoventral diameter of fenestra rotunda | 4.1 | — | 3.4 |
| width of fenestra rotunda | 7.5 | — | 4.3 |
| length of fenestra ovalis | 6.0 | — | 3.7 |
| width of fenestra ovalis | 4.3 | — | 5.4 |
| greatest length of mallear fossa | 7.0 | 5.6 | — |
| greatest length of ventral opening of the facial canal | — | 26.4 | — |
| length from anteroventral angle to tip of lateral tuberosity | 42.7 | 54.4 | — |
| length of the posterior process of the periotic | 139.6 | 114.6 | — |
| bulla | | | |
| greatest length, in lateral view | 94.7 | 94.4 | — |
| greatest length of tympanic cavity | — | 72.8 | — |
| greatest width, in lateral view | — | 74.2 | — |
| greatest height, from the tip of sigmoid process to the ventral-most point in posterior view | — | 65.2 | — |
| height of involucrum at the anteriormost point of the posterior pedicle | 30.8 | 34.0 | — |
| length of anterior lobe, from lateral furrow to the anterior tip of tympanic bulla | — | 42.0 | — |
| greatest length of the posterior pedicle at base | 16.6 | 16.9 | — |

long elliptical posterior pedicle of the tympanic bulla shows a broken surface. The posterior process becomes dorsoventrally high in the middle of the process, and the lateral end is dorsoventrally thin.

## 4.4. Tympanic bulla

The tympanic bullae (figures 6, 8, 11 and 12 and table 1) are longer than their width, but with a swollen anterior portion in ventral view, and dorsoventrally compressed in anteroposterior view.

The anterior lobe of the tympanic bulla is swollen laterally. The ventral to the lateral surface of the anterior lobe has a blunt ridge, which is a part of the main ridge running to the medial surface. Posterior to the anterior lobe, a large triangular sigmoid process projects laterally. Its dorsal most point bends posteriorly (figure 12*a*). The sigmoid process has thick margins, about 8.0 mm (anteroposterior length). Anterior to the sigmoid process, a deep lateral furrow runs transversely and separates the sigmoid process and anterior lobe. Anterior to the sigmoid process, the rim of the outer lip becomes thicker, which is the mallear ridge. The mallear ridge contains structures such as the sulcus for the chorda tympani and fossa for the malleus

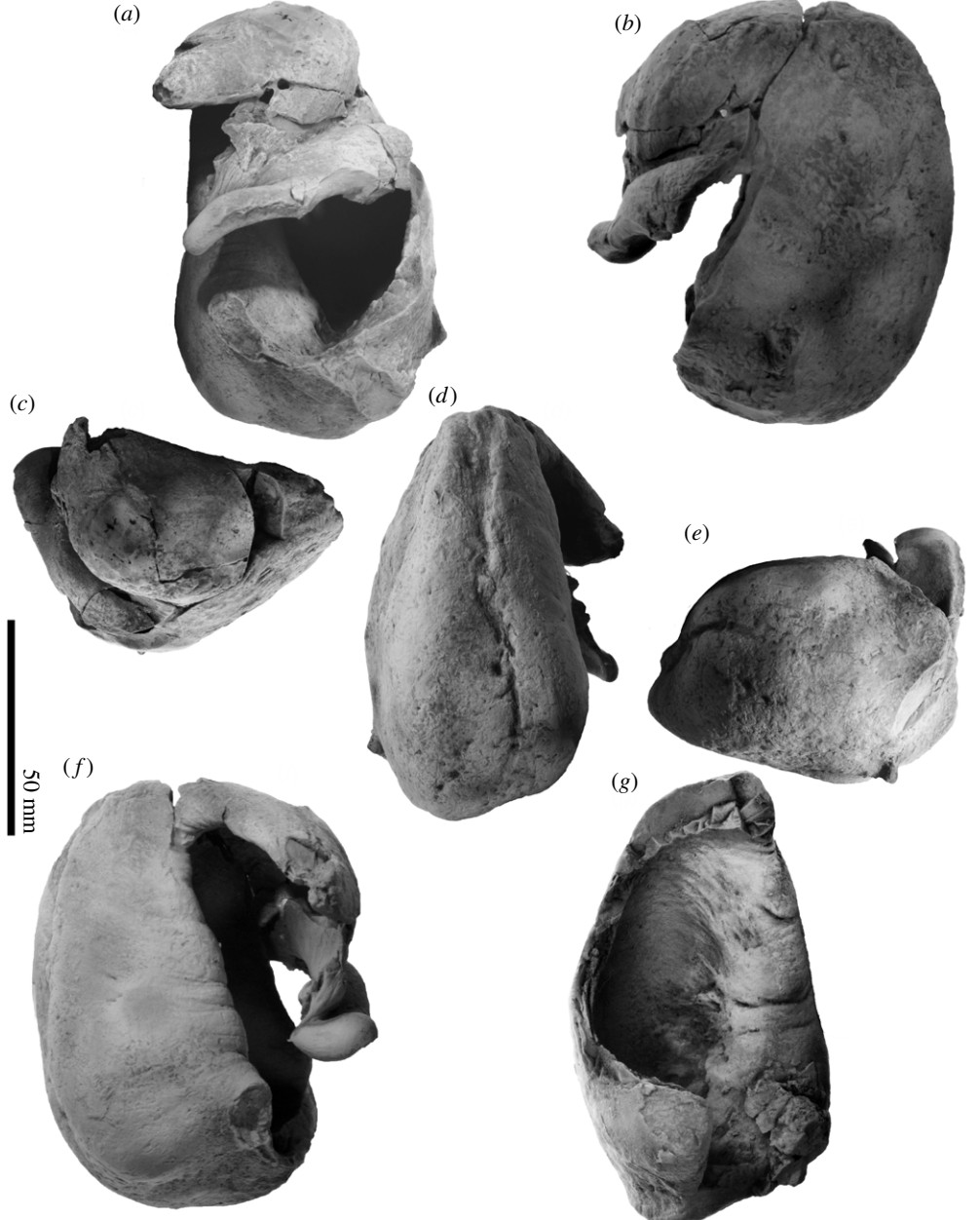

**Figure 11.** Tympanic bullae of FCCP 1049, *A. dosanko*; (*a–f*) right tympanic bulla; (*a*) lateral view; (*b*) ventral view; (*c*) anterior view; (*d*) medial view; (*e*) posterior view; (*f*) dorsal view; (*g*) left tympanic bulla in lateral view.

dorsally. An anteroposteriorly long sulcus for the chorda tympani locates anterior to a long fossa for the malleus. The lateral surface of the mallear ridge has several oblique striae. The involucrum is posteriorly wider (figure 12*g*). Its lateral surface has several transverse grooves. The medial lobe swells medially. A broken base of the posterior pedicle locates at the posterior end of the involucrum on the lateral surface. The broken section of the posterior pedicle is anteroposteriorly long and elliptical (16.6 mm long, 9.0 mm wide). Medial to the posterior pedicle, there is a smooth and low conical process (figure 12*g*). On the medial surface of the tympanic bulla, there are stronger main ridges and weaker involucral ridges running anteroposteriorly, and are not connected to each other. Between them, there is a median furrow, which is deep anteriorly but weak posteriorly. The main ridge runs from the anterior lobe to the posterior end of the tympanic bulla, forming the anteriormost point and also the medial margin of the tympanic bulla.

## 4.5. Mandible

An incomplete right mandible is weakly laterally bowed, especially the anterior part (figure 13 and table 2). The most anterior part has an anteroposteriorly long mental foramen, which also opens anteriorly. The

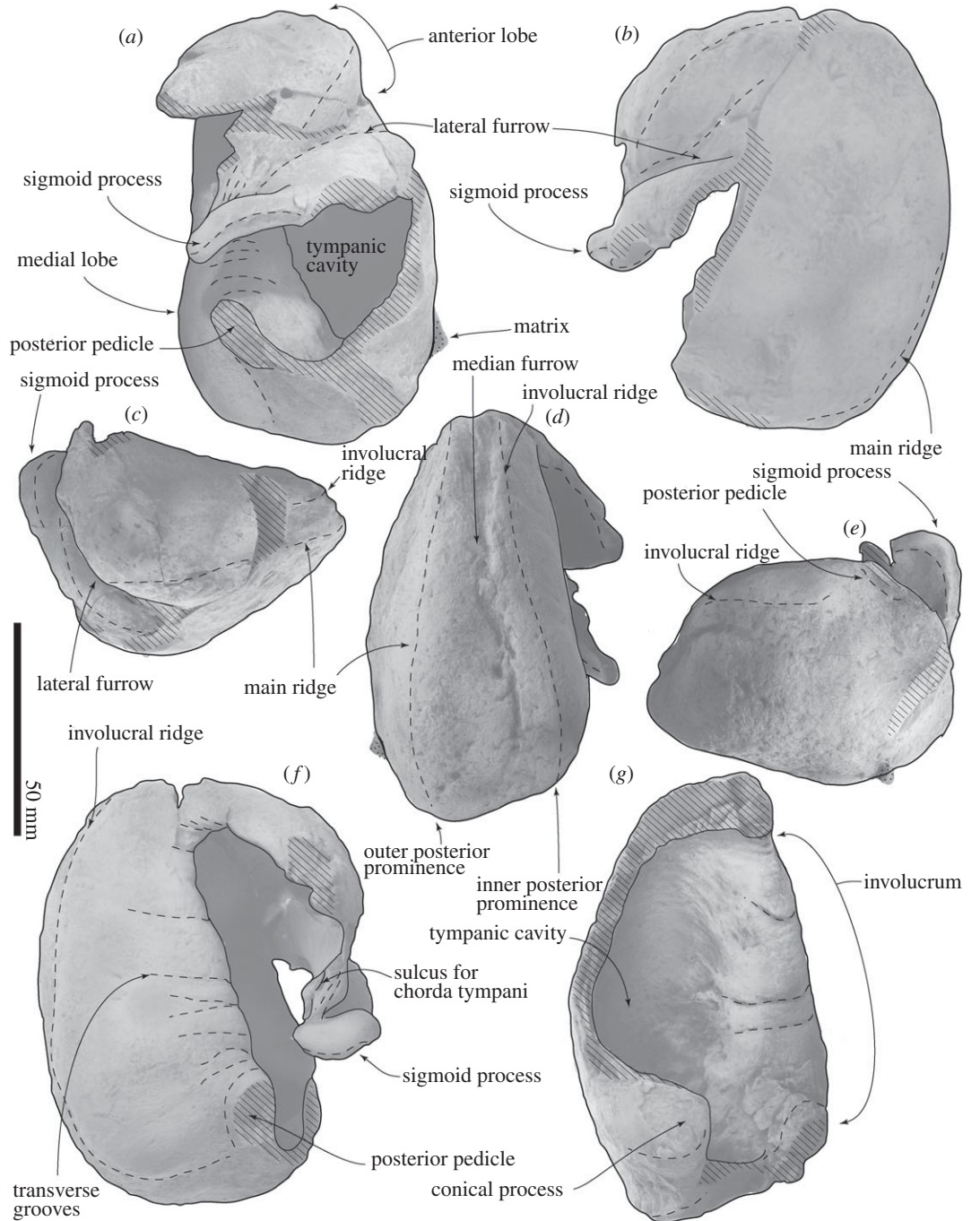

**Figure 12.** Key features of tympanic bullae of FCCP 1049, *A. dosanko*; (*a–f*) right tympanic bulla; (*a*) lateral view; (*b*) ventral view; (*c*) anterior view; (*d*) medial view; (*e*) posterior view; (*f*) dorsal view; (*g*) left tympanic bulla in lateral view.

anterior part of the mental foramen is shallower and wider. Medially, there is a flat area for the mandibular symphysis, which is restricted by a wide symphyseal groove running anteroposteriorly. The symphyseal groove is anteriorly wider and shallower, and its posterior end runs on the medial surface of the mandible. The posterior part preserves a broken coronoid process. The mandibular foramen (about 50 mm high and 40 mm wide at the broken posterior margin) locates dorsally in medial view.

## 4.6. Postcranial skeleton

*Vertebra*. Two thoracic vertebrae (figure 14*a,b*) show a dorsally wider triangular body with a ventral keel, large neural canal and long transverse process. A fossa for the rib locates ventral to the transverse

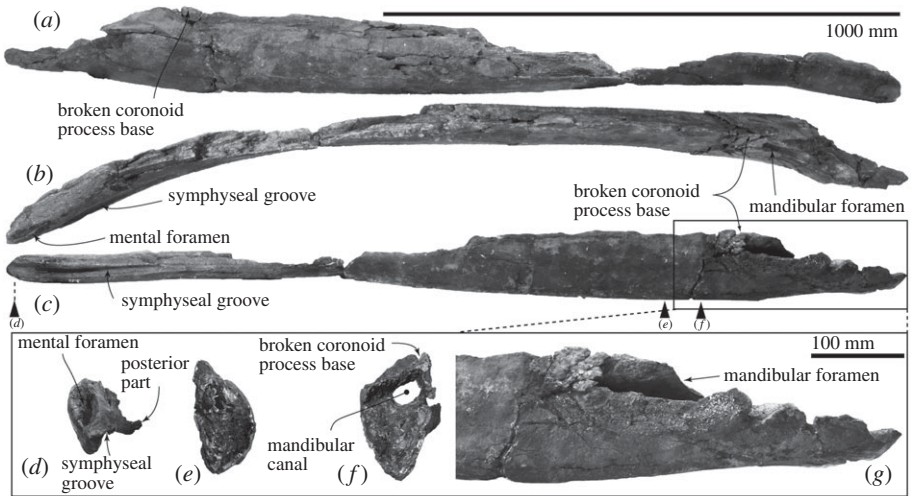

**Figure 13.** Right mandible of FCCP 1049, *A. dosanko*; (*a*) lateral view; (*b*) dorsal view; (*c*) medial view; (*d*–*f*) cross-sections; (*g*) medial view of the posterior end.

process. The vertebral epiphyses are all removed. A caudal vertebra (figure 14*c*) shows the body with fused epiphyses. The supposed anterior articular surface is slightly larger than posterior one (anterior surface: 86.0 mm wide and 93.0 mm high, posterior surface: 80.0 mm wide and 90.0 mm high, length of the body is 71.0 mm). Seven isolated epiphysis (figure 14*d*–*j*) are dorsally wider triangular to laterally wider elliptical. The smallest epiphysis number (figure 14*d*) is 90.0 mm wide and 81.0 mm high. The largest epiphysis number (figure 14*j*) is 131.0 mm wide and 112.0 mm high.

*Presternum.* A preserved part of the presternum shows that symmetrical anterior part is broken and dorsoventral surfaces are worn (figure 14*k* and table 3). A side is convex, which can be identified as the ventral side (figure 14*k*). On the lateral margin, there are a couple of thicker parts, which might be a surface for the ribs.

*Rib.* Five ribs are preserved (figure 15). The rib in figure 15*b* is wide, which might be the most anterior rib. The longest one, figure 15*e*, is about 560.0 mm long and 65.0 mm wide, and its restoration using clay might overestimate its length.

# 5. Discussion

## 5.1. Phylogenetic analysis and comparison

The phylogenetic position of *A. dosanko* gen. et sp. nov. (FCCP 1049) was analysed using the matrix of Buono *et al*. [1], which was derived from the matrix of Marx & Fordyce [37]. This study modified codings of *A. dosanko* with the direct examination (electronic supplementary material, S1–S4) and contains 257 morphological characters and 43 taxa. Percentages of coded data of *A. dosanko* are 46% (includes soft tissue characters), 48% (excludes soft tissue) and 89% for the ear bones.

The matrix was managed using MESQUITE 2.75 [38]. Analysis was performed with TNT v. 1.5 [39]. All of the characters were treated as unweighted and unordered with backbone constraint of extant taxa, based on a topology of the molecular tree [40]. The analysis used New Technology Search with recover minimum length trees = 1000 times.

The phylogenetic analysis shows two shortest trees of 960 steps each. The strict consensus trees (figure 16; electronic supplementary material, S5) are slightly different from that of the equally weighted analysis of Buono *et al*. [1] in that the most basal balaenid is not *P. vexillifer* but *M. parvus*, and *A. dosanko* (*Balaenula* sp. in the previous study) is placed more basal to *B. astensis*. Indeed, the branch lengths of the previous phylogenetic hypothesis around them were relatively low [1].

*Archaeobalaena dosanko* is identical to the all named balaenids. Here, we compare *A. dosanko* with closely related *B. astensis* and *Balaenella brachyrhynus* (table 4). Comparison with *Balaenella brachyrhynus*, *A. dosanko* shows a more strongly curved posterior margin of the supraorbital process in dorsal view, much weaker supramastoid crest, more rounded postorbital process in dorsal view, ventrally almost closed frontal groove by the pre and postorbital ridges, much smaller lateral process

**Figure 14.** Postcranial skeleton of FCCP 1049, *A. dosanko*; (*a,b*) thoracic vertebrae; (*c*) caudal vertebra; (*d–j*) isolated vertebral epiphyses; (*k*) presternum in ventral view.

**Table 2.** Measurements in centimetres of *Archaeobalaena dosanko* (FCCP 1049) skull and mandible following Buono *et al.* [1], and Tanaka & Taruno [36]. (For the skull and mandible, distances are either horizontal or vertical, unless identified as point to point.)

| skull | |
|---|---|
| bizygomatic width | 59.0[a] |
| width of the skull at the level of the exoccipitals | 34.0[a] |
| supraoccipital width anterior to foramen magnum | 32.2[a] |
| supraoccipital width at mid-length | 18.0[a] |
| supraoccipital width at 10 cm to the anterior margin | 12.0[a] |
| supraoccipital length | 38.8 |
| transverse diameter of the left occipital condyle | 7.8 |
| dorsoventral diameter of the left occipital condyle | 9.4[b] |
| transverse diameter of the foramen magnum | 7.2 |
| distance between the lateral border of the left occipital condyle and lateral border of exoccipital | 22.0 |
| width of the occipital condyles plus foramen magnum | 18.8[b] |
| anteroposterior diameter of the supraorbital process of the frontal in the distal edge | 16.0 |
| transverse diameter of the supraorbital process of the frontal | 48.2 |
| anteroposterior diameter of the supraorbital process of the frontal in the constriction | 10.2 |
| width of the optic canal in its medial portion | 0.7 |
| width of the optic canal in its lateral portion | 11.2 |
| length of the optic canal | 40.0[b] |
| orbital length (between preorbital and postorbital process) | 13.0 |
| anteroposterior diameter of the temporal fossa | 27.5 |
| transverse diameter of the temporal fossa | 39.5 |
| *mandible* | |
| length of the mandible, as preserved in a straight line | 180.0 |
| height of the mandible, from coronoid process to ventral margin | 15.5[b] |
| maximum preserved height of mandible | 15.5[b] |
| maximum preserved width of mandible | 74.0 |
| height of the body at just anterior to coronoid process | 15.3 |
| width of the body at just anterior to coronoid process | 74.0 |

[a]Measurements taken from only one side.
[b]An incomplete measurement because of erosion.

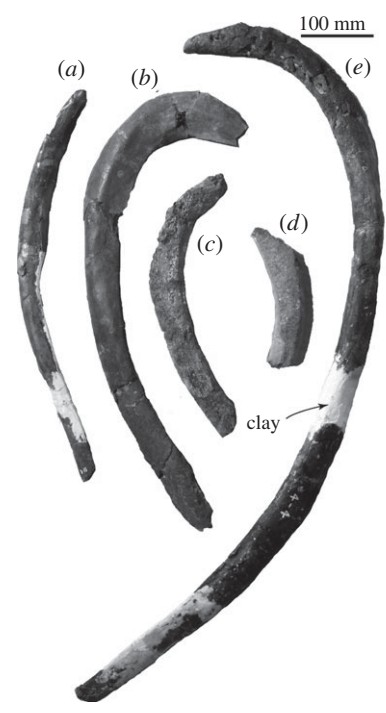

**Figure 15.** Ribs of FCCP 1049, *A. dosanko* (*a–c*) right ribs; (*d,e*) left ribs.

**Table 3.** Measurements in millimetres of *Archaeobalaena dosanko* (FCCP 1049) vertebrae and presternum. (Measurements are rounded to the nearest 0.5 mm. Distances are either horizontal or vertical, unless identified as point to point.)

| *thoracic vertebra in* figure 14*a* | |
| --- | --- |
| maximum preserved length | 77.0 |
| maximum preserved height | 270.0 |
| maximum preserved width | 191.0[a] |
| length of the neural spine | 48.0 |
| height of the neural spine | 115.0 |
| length of the body | 52.0 |
| width of the body anteriorly | 122.0 |
| height of the body anteriorly | 92.0 |
| width of the body posteriorly | 144.0 |
| height of the body posteriorly | 90.0 |
| *thoracic vertebra in* figure 14*b* | |
| maximum preserved length | 86.0 |
| maximum preserved height | 172.0 |
| maximum preserved width | 237.0 |
| length of the body | 49.0 |
| width of the body anteriorly | 125.0 |
| height of the body anteriorly | 77.0 |
| width of the body posteriorly | 132.0 |
| height of the body posteriorly | 92.0 |
| *sternum* | |
| maximum preserved length | 118.0 |
| maximum preserved height | 18.0 |
| maximum preserved width | 83.0 |

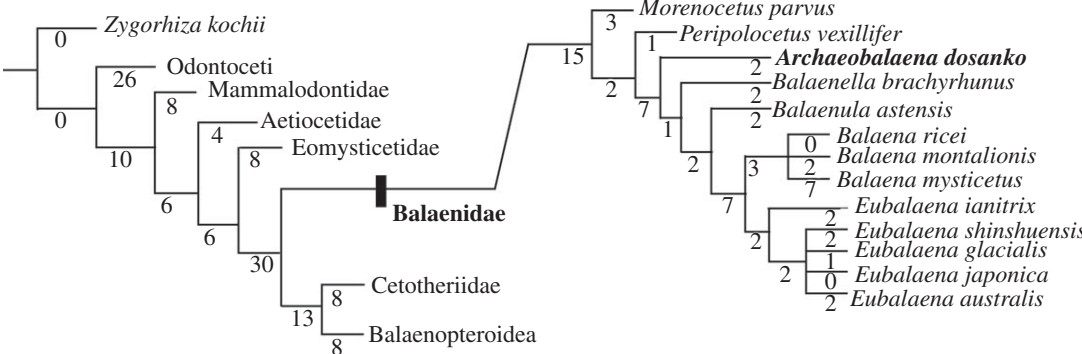

**Figure 16.** Strict consensus tree of two most parsimonious trees to show the phylogenetic position of *A. dosanko* gen. et sp. nov. The most parsimonious trees were 960 steps. The clades Aetiocetidae, Eomysticetidae, Balaenidae, Cetotheriidae and a clade comprising *Isanacetus*, *Parietobalaena* and related taxa are collapsed. (See the electronic supplementary material, file S5 for cladogram with all taxa shown.)

**Table 4.** Comparison between *Archaeobalaena dosanko* (FCCP 1049) and related fossil balaenids using previous publications [6,41].

|  | Archaeobalaena dosanko | Balaenula balaenopsis | Balaenula astensis | Balaenella brachyrhynus |
|---|---|---|---|---|
| pre and postorbital processes | weak | strongly project ventrally | strongly project ventrally | weak |
| lateral margin of supraorbital process in dorsal view | straight | strongly excavated | straight | straight |
| supraorbital process | slender | slender | robust | slender |
| orbitotemporal crest | weak, only on medial part of supraorbital process | strong | strong | weak, only on medial part of supraorbital process |
| supramastoid crest | weaker | stronger | weaker | weaker |
| zygomatic process | slender | robust | robust | — |
| distinct tubercle at junction of parieto-squamosal suture and supraoccipital (character 82) | absent | — | absent | present |
| lateral margin of the nuchal crest | laterally projected | — | straight | laterally projected |
| anterior supraoccipital fossa | absent | — | present | absent |
| posterior supraoccipital fossa position | just anterior to occipital condyles | — | more anterior | — |

of the periotic, thinner medial margin of the fenestra rotunda, much weaker outer posterior prominence of the tympanic bulla, much stronger inner posterior prominence and more rounded anteromedial angle of the tympanic bulla in ventral and dorsal views. Of note, *Balaenella brachyrhynus* does not preserve the anterior tip of the zygomatic process.

Comparison with *B. astensis*, *A. dosanko* shows a more slender supraorbital process, no tubercle at the junction of the parieto-squamosal suture and supraoccipital (character 82), lateral projection of the nuchal crest and more posteriorly located posterior supraoccipital fossa.

As is mentioned above, the type species of the genus *Balaenula*: the holotype of *B. balaenopsis* is difficult to recognize as an individual [1]. Indeed, the long and triangular anterior process of the periotic on the type specimen probably does not belong to the Balaenidae, but Balaenopteroidea. Based on illustrations of previous studies [2,41], the periotic seems having some balaenopteriid diagnoses such as a narrow and triangular anterior process [14,42], but lacks other diagnoses for the family such as a well-defined fossa for the malleus [14] and transversely elongated pars cochlearis [43]. Thus, it might be a stem balaenopteroid. Its overall shape in ventral view is similar to that of *Tiphyocetus temblorensis* of Kellogg [44] by having a slander anterior process and globular pars cochlearis, but morphologies of the internal acoustic meatus and lateral tuberosities are different.

However, the holotype skull of *B. balaenopsis* belongs to the Balaenidae, and can we compare here. In comparison with *B. astensis* + *B. balaenopsis*, *A. dosanko* shows weaker pre and postorbital processes in lateral view, and weaker orbitotemporal crest only on the medial part of the supraorbital process. In comparison with *B. balaenopsis*, *A. dosanko* shows a more or less straight lateral margin of the supraorbital process, and much weaker supramastoid crest. Some *Balaenula* sp. specimens from the late Miocene to early Pliocene, and Late Pliocene of California have been mentioned, but not photographed and illustrated [45].

## 5.2. The supramastoid crest

The supramastoid crest of balaenids are generally developed, but morphological change of the supramastoid crest cannot be thought simple, like developing from incipient to large through balaenid evolution. The oldest known balaenid *M. parvus* shows a small supramastoid crest on the holotype, but just a ridge along the dorsal surface of the zygomatic process on the referred specimen (MPL 5-15) [1].

The modern balaenids show varied conditions of the supramastoid crest. *Balaena mysticetus* shows both small (a juvenile individual described by Nishiwaki & Kasuya [46]) and well developed (USNM 257513, see fig. 5 of Bisconti [2]) conditions, as Field *et al.* [47] mentioned. *Eubalaena australis* shows small ones (a young individual in Best [48] and USNM 26712, see fig. 4 of Bisconti [2]). Possibly, the development of the supramastoid crest depends on the ontogenetic stages. Some fossil species belonging to the genus *Balaena* and *Eubalaena* such as *Eubalaena ianitrix* also have small ridges [5,49], but *B. montalionis* and *Eubalaena* sp. (IMNH 9598) have large ridges [8].

As mentioned above, having a low and rounded supramastoid crest was considered a diagnosis of the genus *Balaenula*. However, early Pliocene balaenids (*A. dosanko*, *Balaenula* spp. and *Balaenella brachyrhynus*) have relatively large supramastoid crests among the Balaenidae. Among them, the conditions of the supramastoid crest are varied as compared with above (see also table 4), especially *B. balaenopsis* has a larger and dorsally strongly projected supramastoid crest than those of *A. dosanko*, *B. astensis* and *Balaenella brachyrhynus*.

The rounded and low supramastoid crest had emerged already among the Balaenidae during the early Miocene as an incipient condition, then enlarged by the early Pliocene, and some kept the condition or others had the crest as a smaller secondary. In short, the morphological change of the supramastoid crest is not simple. The supramastoid crest is attached to the temporal fascia [50,51]. Thus, these conditions of the supramastoid crest might be related to the size and/or orientation of the temporal muscle. At least, from *M. parvus* to Pliocene balaenids (including *A. dosanko*), the width of the skull was increased twice, which was the result of lateral elongation of the supraorbital and zygomatic processes, and a mediolaterally expanded temporal fossa. This size and shape change of the skull is effected to shape the modification of the temporal muscle to control the mandibles effectively. However, the possibility of anteroposterior expansion of the temporal muscle might be limited owing to the elongation of the rostrum and whole body proportion. Thus, the temporal muscle earned a larger physically effective attachment point: the supramastoid crest. Many other factors such as width of the supraorbital process, degree of telescoping, arching of the rostrum and orientation of the skull can be thought of. However, the number of rostrum records is not enough to consider.

# 6. Conclusion

*Archaeobalaena dosanko* gen. et sp. nov. (FCCP 1049) represents an archaic balaenid from the early Pliocene, Zanclean (3.5–5.2 Ma) lower part of the Chippubetsu Formation, Hokkaido, Japan. *Archaeobalaena dosanko* is distinguishable from another balaenids by having a deep promontorial groove of the pars cochlearis of the

periotic. *Archaeobalaena dosanko* can be differentiated from other balaenids, except *M. parvus* by having a slender zygomatic process, and posteriorly oriented postorbital process in dorsal view. The result of phylogenetic analysis places *A. dosanko* more basal to *B. astensis*. *Archaeobalaena dosanko* adds detailed skull, periotic and bulla morphologies for the earlier balaenids.

Data accessibility. The data for the phylogenetic analysis are available as electronic supplementary material. The LSID for this publication is: urn:lsid:zoobank.org:pub:322CDB25-4971-46A1-B40F-6200A8CCAF03.

Authors' contributions. Y.T. carried out the phylogenetic analysis and wrote the paper. H.F. and M.K. revised the manuscript.

Competing interests. The authors declare that they have no competing interests.

Funding. There is no funding for this paper.

Acknowledgements. We thank J. Takahashi, N. Mita and others for discovering, and to 100s of people for digging up the study material between 1978 and 1980. We also thank T. Kimura (Gunma Museum of Natural History) and M. R. Buono (CCT CONICET-CENPAT) for giving constructive comments, and staff of Fukagawa City Board of Education for access to the specimen.

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
