## [Reviewer comments · Royal Society Open Science]

Review History

RSOS-192182.R0 (Original submission)

Review form: Reviewer 1 (Toshiyuki Kimura)

Is the manuscript scientifically sound in its present form?

Yes

Are the interpretations and conclusions justified by the results?

Yes

Is the language acceptable?

Yes

Do you have any ethical concerns with this paper?

No

Have you any concerns about statistical analyses in this paper?

No

Recommendation?

Accept with minor revision (please list in comments)

Comments to the Author(s)

The manuscript "A new member of fossil balaenid (Mysticeti, Cetacea) from the early Pliocene of Hokkaido, Japan" by Tanaka et al. represents an interesting addition to our knowledge of the evolution of the Balaenidae. The author describes a Pliocene balaenid as new species "Balaenula" dosanko. Since the scarcity of the fossil record of the balaenids, I suspect this paper will be widely received by people interested in the evolution of cetaceans. I annotated the pdf file of the manuscript for the revision. Please find attached file (Appendix A).

I have a few comments and suggestions that I think it will improve the paper.

- 1) There are several typos in the text. Please double check.
- 2) The authors preliminary diagnosed the specimen (FCCP 1049) as "Balaenula" dosanko, sp. nov. The tentative generic combination is allowed by ICZN. However, in this paper, the authors clearly mention that FCCP 1049 is differed from the other balaenid (e.g., page4 lines 20-25). I think that the authors should explain on why they do not describe FCCP 1049 as new genus.
- 3) The authors preliminary diagnosed FCCP 1049 as genus "Balaenula". But, they do not adequately explain on the reason for their (preliminary) decision. Even if it is a preliminary decision, I think the authors should justify their decision based on the morphological characters found in the specimen.
- 4) The authors mentioned that ["Balaenula" dosanko is hard to determine the genus belonged. Because the result of this phylogenetic analysis show clade as an unsolved polytomy.] (page 7. lines 48-51). I think that the taxon is not defined by the topology of the tree itself, but the taxon is defined by the characters (e.g., ICZN Art. 13.1.1: "every new name [of taxon] must be accompanied by a description or definition that states in words characters that are purported to differentiate the taxon). The authors clearly mention that FCCP 1049 is differed from the other balaenid (e.g., page4 lines 20-25). I would like to recommend the authors to describe the specimen as new genus.

Toshiyuki Kimura
Gunma Museum of Natural History

Review form: Reviewer 2 (Mónica Buono)

Is the manuscript scientifically sound in its present form?

Yes

Are the interpretations and conclusions justified by the results?

No

Is the language acceptable?

No

Do you have any ethical concerns with this paper?

No

Have you any concerns about statistical analyses in this paper?

No

Recommendation?

Major revision is needed (please make suggestions in comments)

Comments to the Author(s)

General comments:

The manuscript "A new member of fossil balaenid (Mysticeti, Cetacea) from the early Pliocene of Hokkaido, Japan" re-describes a species of Balaenidae from the early Pliocene of Japan. In this work, the authors perform a re-description of the specimen and revise its taxonomic status. It is an important contribution to the knowledge of taxonomic diversity of extinct balaenids, especially from the important but slightly misleading Japan fossil record. The description is based on an incomplete skull with ear bones preserved, and some postcranial elements. The structure and organization of the manuscript are good. I am not an English speaker, but I strongly suggest to revise the English grammar because many sections of the paper are hard to understand. However, I have some main concerns that are outlined in detail below and in the attached file as well, which I hope the authors take into consideration. For that reason, my decision is a major revision.

My main concerns are in the following sections of the manuscript (see Appendices B & C):

1) Introduction: this section needs to be improved in order to be more informative for readers.

Some references should be added and there are many comments and corrections I have suggested in the pdf file.

2) Diagnosis and description:

-Diagnosis: some characters should be revised considering they are not apomorphic for this taxon (i.e. morphology of zygomatic process). I suggest that the authors first work in a differential diagnosis in order to better determine if the specimen is a new genus or if it could be maintained inside of *Balaenula*. I think that this would be more important than providing a new species name. There is enough anatomical evidence to differentiate this specimen from *Morenocetus*, *Balaena*, *Eubalaena* and *Balaenella*, considering not only skull but also the tympano-periotic morphology (I suggest the authors should focus on that). Then, continue to present if there is enough evidence to nominate a new species. More comments in the pdf file.

-Description: I have made many suggestions in the description section, some of them include the revision of some anatomical structures which should be taken into account by the authors.

3) Phylogenetic analysis:

My main concern with this point is that authors perform only one analysis, using a molecular constraint, which shows "*Balaenula* dosanko" in a polytomy. Why did the authors not explore other types of phylogenetic analysis? For example, without the molecular constraint (and between equal and implied weights?). Besides, the type species *Balaenula balaenopsis* should be added to the analysis, especially considering the authors are testing the validity of a *Balaenula* species's. Please also check my comments in the supplementary file S4 modified coding.

4) Discussion.

One of the main problems that I saw in the paper is the taxonomic status of the specimen studied (FCCP 1049). Many sections of the manuscript pointed out that the assignation to genus *Balaenula* is not well supported (for example, there are no characters supporting its diagnosis and also the taxon is recovered in a polytomy with *B. astensis* and *Balaenella*). The re-diagnosis provided by Bisconti (2003) included other characters not discussed by the authors. Besides, the phylogenetic analysis did not include the type species of *Balaenula*. I understand the status of this taxon is conflictive, however, is hard to really test the validity of this species and its assignation to *Balaenula* if the type species is not included in the analysis. Why did the authors decide to maintain this assignation?

I recommend to the authors to revise the phylogenetic analysis, include *B. balaenopsis* (at least with the materials which are more reliable such as the skull and skeleton) and include deeper comparisons with both taxa: *B. balaenopsis* and *B. astensis*. It should be noted that *Balaenula astensis* morphology should be interpreted with caution because the specimen exhibits many juvenile characters. In my opinion, skull characters are not fully useful to diagnose *Balaenula*; periotic and bulla morphology should be further considered, in particular between "*Balaenula* dosanko" and *B. astensis*. Thorough comparisons of the tympano-periotic morphology should be made in order to discuss differences or similarities, which could better support (or not) the assignation of FCCP 1049 to *Balaenula*.

Finally, I suggest to further expand the discussion of the evolution of supramastoid crest in balaenids (see comments on the pdf).

Figures:

Generally, the figures are fine, and of high quality. However, there are spelling mistakes to be corrected (see pdf). The scales in the figures should be consistently placed in all figures in the same place.

Figure 6: it needs revision because there are structures (premaxilla vs maxilla) that are not consistently interpreted as in the description section of the manuscript.

Tables:

Please check the format of the tables (different font and font size are observed). Besides, many words appear to be cut.

Mónica R. Buono, PhD
IPGP- CCT CONICET-CENPAT
Puerto Madryn-Chubut-Argentina
Tel: (54) (280) 4883184 ext.1320
buono@cenpat-conicet.gob.ar
m.buono8@gmail.com

Decision letter (RSOS-192182.R0)

27-Feb-2020

Dear Dr Tanaka

On behalf of the Editors, I am pleased to inform you that your Manuscript RSOS-192182 entitled "A new member of fossil balaenid (Mysticeti, Cetacea) from the early Pliocene of Hokkaido, Japan" has been accepted for publication in Royal Society Open Science subject to minor revision in accordance with the referee suggestions. Please find the referees' comments at the end of this email.

The reviewers and handling editors have recommended publication, but also suggest some minor revisions to your manuscript. Therefore, I invite you to respond to the comments and revise your manuscript. Pay special attention to the use of English and correct any typographical errors.

- Ethics statement

- Data accessibility

If you wish to submit your supporting data or code to Dryad (<http://datadryad.org/>), or modify your current submission to dryad, please use the following link:
<http://datadryad.org/submit?journalID=RSOS&manu=RSOS-192182>

- **Competing interests**

- **Authors' contributions**

- **Acknowledgements**

- **Funding statement**

Because the schedule for publication is very tight, it is a condition of publication that you submit the revised version of your manuscript before 07-Mar-2020. Please note that the revision deadline will expire at 00.00am on this date. If you do not think you will be able to meet this date please let me know immediately.

If your manuscript is newly submitted and subsequently accepted for publication, you will be asked to pay the article processing charge, unless you request a waiver and this is approved by Royal Society Publishing. You can find out more about the charges at <https://royalsocietypublishing.org/rsos/charges>. Should you have any queries, please contact openscience@royalsociety.org.

on behalf of Professor Rachel Wood (Associate Editor) and Jon Blundy (Subject Editor)
openscience@royalsociety.org

Reviewer comments to Author:

Reviewer: 1

Comments to the Author(s)

The manuscript "A new member of fossil balaenid (Mysticeti, Cetacea) from the early Pliocene of Hokkaido, Japan" by Tanaka et al. represents an interesting addition to our knowledge of the evolution of the Balaenidae. The author describes a Pliocene balaenid as new species "Balaenula" dosanko. Since the scarcity of the fossil record of the balaenids, I suspect this paper will be widely received by people interested in the evolution of cetaceans. I annotated the pdf file of the manuscript for the revision. Please find attached file.

I have a few comments and suggestions that I think it will improve the paper.

- 1) There are several typos in the text. Please double check.
- 2) The authors preliminary diagnosed the specimen (FCCP 1049) as "Balaenula" dosanko, sp. nov. The tentative generic combination is allowed by ICZN. However, in this paper, the authors clearly mention that FCCP 1049 is differed from the other balaenid (e.g., page4 lines 20-25). I think that the authors should explain on why they do not describe FCCP 1049 as new genus.
- 3) The authors preliminary diagnosed FCCP 1049 as genus "Balaenula". But, they do not adequately explain on the reason for their (preliminary) decision. Even if it is a preliminary decision, I think the authors should justify their decision based on the morphological characters found in the specimen.
- 4) The authors mentioned that ["Balaenula" dosanko is hard to determine the genus belonged. Because the result of this phylogenetic analysis show clade as an unsolved polytomy.] (page 7. lines 48-51). I think that the taxon is not defined by the topology of the tree itself, but the taxon is defined by the characters (e.g., ICZN Art. 13.1.1: "every new name [of taxon] must be accompanied by a description or definition that states in words characters that are purported to differentiate the taxon). The authors clearly mention that FCCP 1049 is differed from the other balaenid (e.g., page4 lines 20-25). I would like to recommend the authors to describe the specimen as new genus.

Toshiyuki Kimura

Gunma Museum of Natural History

Reviewer: 2

Comments to the Author(s)

General comments:

The manuscript "A new member of fossil balaenid (Mysticeti, Cetacea) from the early Pliocene of Hokkaido, Japan" re-describes a species of Balaenidae from the early Pliocene of Japan. In this work, the authors perform a re-description of the specimen and revise its taxonomic status. It is an important contribution to the knowledge of taxonomic diversity of extinct balaenids, especially from the important but slightly misleading Japan fossil record. The description is based on an incomplete skull with ear bones preserved, and some postcranial elements. The structure and organization of the manuscript are good. I am not an English speaker, but I strongly suggest to revise the English grammar because many sections of the paper are hard to understand. However, I have some main concerns that are outlined in detail below and in the attached file as well, which I hope the authors take into consideration. For that reason, my decision is a major revision.

My main concerns are in the following sections of the manuscript:

1) Introduction: this section needs to be improved in order to be more informative for readers. Some references should be added and there are many comments and corrections I have suggested in the pdf file.

2) Diagnosis and description:

-Diagnosis: some characters should be revised considering they are not apomorphic for this taxon (i.e. morphology of zygomatic process). I suggest that the authors first work in a differential diagnosis in order to better determine if the specimen is a new genus or if it could be maintained inside of *Balaenula*. I think that this would be more important than providing a new species name. There is enough anatomical evidence to differentiate this specimen from *Morenocetus*, *Balaena*, *Eubalaena* and *Balaenella*, considering not only skull but also the tympano-periotic morphology (I suggest the authors should focus on that). Then, continue to present if there is enough evidence to nominate a new species. More comments in the pdf file.

-Description: I have made many suggestions in the description section, some of them include the revision of some anatomical structures which should be taken into account by the authors.

3) Phylogenetic analysis:

My main concern with this point is that authors perform only one analysis, using a molecular constraint, which shows "*Balaenula*" *dosanko* in a polytomy. Why did the authors not explore other types of phylogenetic analysis? For example, without the molecular constraint (and between equal and implied weights?). Besides, the type species *Balaenula balaenopsis* should be added to the analysis, especially considering the authors are testing the validity of a *Balaenula* species's. Please also check my comments in the supplementary file S4 modified coding.

4) Discussion.

One of the main problems that I saw in the paper is the taxonomic status of the specimen studied (FCCP 1049). Many sections of the manuscript pointed out that the assignation to genus *Balaenula* is not well supported (for example, there are no characters supporting its diagnosis and also the taxon is recovered in a polytomy with *B. astensis* and *Balaenella*). The re-diagnosis provided by Bisconti (2003) included other characters not discussed by the authors. Besides, the phylogenetic analysis did not include the type species of *Balaenula*. I understand the status of this taxon is conflictive, however, is hard to really test the validity of this species and its assignation to *Balaenula* if the type species is not included in the analysis. Why did the authors decide to maintain this assignation?

I recommend to the authors to revise the phylogenetic analysis, include *B. balaenopsis* (at least with the materials which are more reliable such as the skull and skeleton) and include deeper comparisons with both taxa: *B. balaenopsis* and *B. astensis*. It should be noted that *Balaenula astensis* morphology should be interpreted with caution because the specimen exhibits many juvenile characters. In my opinion, skull characters are not fully useful to diagnose *Balaenula*; periotic and bulla morphology should be further considered, in particular between "*Balaenula*" *dosanko* and *B. astensis*. Thorough comparisons of the tympano-periotic morphology should be made in order to discuss differences or similarities, which could better support (or not) the assignation of FCCP 1049 to *Balaenula*.

Finally, I suggest to further expand the discussion of the evolution of supramastoid crest in balaenids (see comments on the pdf).

Figures:

Generally, the figures are fine, and of high quality. However, there are spelling mistakes to be corrected (see pdf). The scales in the figures should be consistently placed in all figures in the same place.

Figure 6: it needs revision because there are structures (premaxilla vs maxilla) that are not consistently interpreted as in the description section of the manuscript.

Tables:

Please check the format of the tables (different font and font size are observed). Besides, many words appear to be cut.

buono@cenpat-conicet.gob.ar
m.buono8@gmail.com

Author's Response to Decision Letter for (RSOS-192182.R0)

See Appendix D.

Decision letter (RSOS-192182.R1)

19-Mar-2020

Dear Dr Tanaka,

It is a pleasure to accept your manuscript entitled "A new member of fossil balaenid (Mysticeti, Cetacea) from the early Pliocene of Hokkaido, Japan" in its current form for publication in Royal Society Open Science. The comments of the reviewer(s) who reviewed your manuscript are included at the foot of this letter.

on behalf of Professor Rachel Wood (Associate Editor) and Jon Blundy (Subject Editor)
openscience@royalsociety.org

Appendix A**ROYAL SOCIETY
OPEN SCIENCE****A new member of fossil balaenid (Mysticeti, Cetacea) from
the early Pliocene of Hokkaido, Japan**

Journal:	Royal Society Open Science
Manuscript ID	RSOS-192182
Article Type:	Research
Date Submitted by the Author:	18-Dec-2019
Complete List of Authors:	Tanaka, Yoshimasa; Osaka Museum of Natural History; Hokkaido University Museum, Division of Academic Resources and Specimens; Numata Fossil Museum Furusawa, Hitoshi; Sapporo Museum Activity Center Kimura, Masaichi; Hokkaido University of Education
Subject:	Palaeontology < EARTH SCIENCES
Keywords:	Balaenidae, New species, Zanclean, Balaenula balaenopsis, supramastoid crest
Subject Category:	Earth science

**Author-supplied statements**

Relevant information will appear here if provided.

**Ethics**

*Does your article include research that required ethical approval or permits?:*

This article does not present research with ethical considerations

*Statement (if applicable):*

CUST_IF_YES_ETHICS :No data available.

**Data**

*It is a condition of publication that data, code and materials supporting your paper are made publicly*
*available. Does your paper present new data?:*

Yes

*Statement (if applicable):*

The data for the phylogenetic analysis is available as electric supplementaries.

**Conflict of interest**

I/We declare we have no competing interests

*Statement (if applicable):*

CUST_STATE_CONFLICT :No data available.

**Authors' contributions**

This paper has multiple authors and our individual contributions were as below

*Statement (if applicable):*

Y. T. carried out the phylogenetic analysis and write the paper. H. F. and M. K. revised the
manuscript.

<Title page>

1) A new member of fossil balaenid (Mysticeti, Cetacea) from the early Pliocene of
Hokkaido, Japan

2) Yoshihiro Tanaka^{1,2,3} Hitoshi Furusawa⁴ Masaichi Kimura^{3,5}

¹ Osaka Museum of Natural History, Nagai Park 1-23, Higashi-Sumiyoshi-ku, Osaka, 546-
0034, Japan

tanaka@mus-nh.city.osaka.jp

² Division of Academic Resources and Specimens, Hokkaido University Museum

³ Numata Fossil Museum

⁴ Sapporo Museum Activity Center

hitoshi.furusawa@city.sapporo.jp

⁵ Hokkaido University of Education

mkimura1313@yahoo.co.jp

3) Corresponding author: Yoshihiro TANAKA

Osaka Museum of Natural History

Nagai Park 1-23, Higashi-Sumiyoshi-ku, Osaka, 546-0034, JAPAN

Tel: +81-(0)6-6697-6222

E-mail: tanaka@mus-nh.city.osaka.jp

Abstract

The family Balaenidae includes two genus and four extant species, and at least eight extinct
species. The oldest known record of the members of the Balaenidae is known from the early
Miocene, but still need more early members of the family to provide better phylogenetic
hypotheses. FCCP 1049 from lower part of the Chippubetsu Formation, Fukagawa Group
(3.5 to 5.2 Ma, Zanclean, early Pliocene) was preliminary described and identified as
*Balaenula* sp. by Furusawa and Kimura in 1982. Later works discussed that FCCP 1049 is
different from the genus, and is placed in different clade from *Balaenula astensis*. In this
study, FCCP 1049 is re-described and named as "*Balaenula*" *dosanko* sp. nov. FCCP 1049
is distinguishable from another balaenids by having a slender zygomatic process, and deep
promontorial groove of the pars cochlearis of the periotic. The result of phylogenetic analysis
places "*Balaenula*" *dosanko* among an unsolved polytomy of *Balaenula astensis* +
*Balaenella brachyrhynchus* + crown clade. "*Balaenula*" *dosanko* is preliminary belonged to
the genus "*Balaenula*", in this study.

ADDITIONAL KEYWORDS: Balaenidae - new species - Zanclean - *Balaenula balaenopsis* – supramastoid crest -

1. Introduction

A baleen whale group, the family Balaenidae includes two genus and four extant species, and at least eight extinct species. The modern balaenid body length reach 17 to 20 m [1]. The oldest known nominal species of the members of the Balaenidae is known from the early Miocene [2], and its body length was estimated as 4.8 to 6.2 m [3]. Our knowledge of balaenid diversity and gigantism are growing [2–6], but still need more early members of the family to hypothesize their phylogeny.

A Pliocene genus *Balaenula* had been treated as “a taxonomical basket, where all the small-sized balaenids were put” in history [7]. The first species of the genus *Balaenula balaenopsis* from Antwerp was established by Van Beneden [8]. But, the holotype is doubtful to recognize as an individual [2]. Supposed similar case of establishing new species with mixed individuals was happened on a cetotheriid, *Herpetocetus scaldiensis* of Van Beneden [8], which was summarized by Deméré et al. [9]. The second species, *Balaenula astensis* from the late early Pliocene of Villafranca d’Asti was established by Trevisan [10] and re-described by Bisconti [11].

Furusawa and Kimura [12] preliminary identified FCCP 1103 from an early Pliocene sediment in Hokkaido, Japan as *Balaenulla* sp. based on showing a low triangle shaped occipital with a depressed at the center, an acute angle of the nuchal crest against the plain in posterior view, and a rounded exoccipital placing below to the ventral margin of the occipital condyle. The study mentioned that these features can be seen only on *Balaenula balaenopsis*. Bisconti [11] mentioned that FCCP 1049 is different from *Balaenula* based on an extended temporal fossa (in page 47). One of the most recent phylogeny works found that FCCP 1049 does not form a clade with *Balaenula astensis* [2]. To expand diversity and morphological information for understanding the early balaenid evolution, we update identification of FCCP 1049 and its geological age, and also re-describe in this study.

2. Material and methods

Morphological terminology follows Mead and Fordyce [13].

2.1. Institutional abbreviations

FCCP, Fukagawa City Cultural Properties, Hokkaido, Japan. IMNH, Icelandic Museum of Natural History. MLP, Museo de La Plata, La Plata, Argentina. USNM, National Museum of Natural History, Smithsonian Institution, Washington D.C.

3. Systematic paleontology

Cetacea Brisson, 1762

Neoceti Fordyce and de Muizon, 2001

Mysticeti Gray, 1864

Chaeomysticeti Mitchell, 1989

Balaenidae Gray, 1825

“*Balaenula*” van Beneden, 1872

Type species. Balaenula balaenopsis

“*Balaenula*” *dosanko* sp. nov

LSID

Holotype. FCCP 1049, including the premaxilla, maxilla, frontal, parietal, squamosal, exoccipital, supraoccipital, periotics, tympanic bullae, right mandible, two thoracic vertebrae, a caudalvertebra, presternum and five ribs. *FCCP 1049 was previously registered as HUES (Hokkaido University of Education Sapporo Campus) 100003.*

Furusawa and Kimura [12] preliminary identified as *Balaenulla* sp. In this study, FCCP 1049 is still preliminary belonged to the genus *Balaenula*, because that its phylogenetic position is among an unsolved polytomy. See more in discussion.

Locality and horizon. FCCP 1049 was found at a river bed of Tadoshi River in Fukagawa

City, Hokkaido, Japan by J. Takahashi, N. Mita and others in 22 September 1978 and dug up in 1979 and 1980 by over 100 of people [19]: Latitude 43°47'36.31"N, longitude 142°5'10.00"E (Figure 1).

FCCP 1049 was found from lower part of the Chippubetsu Formation, Fukagawa Group, especially lower to so-called T1 tuff layer [20]). Regarding the study, above T1 tuff layer, there is T2 tuff layer, which is correlated to S1 tuff of the Takikawa Formation. The age of S1 tuff is 4.1 ± 0.6 Ma [21]. T1 tuff possibly distributes above Ops tuff of upper part of the Horokaoshirarika Formation [20]. The age of Ops tuff is 4.5 ± 0.7 Ma. Thus, the age of FCCP 1049 can be taken as 3.5 to 5.2 Ma (Zanclean, early Pliocene) with wider ranges. Very near from the locality (about 10 to 20 km away), *Numataphocoena yamashitai* and *Herpetocetinae* gen. et sp. indet. and some Mysticeti indet. materials from the early Pliocene (about 4.5 to 3.5 Ma) possibly be cetaceans in the same age [22–26]. From the same area, there are some late Miocene mysticeti reports: type specimen of *Miobalaenoptera numataensis* (6.5 to 6.8 Ma) and a referred specimen *Herpetocetus* sp. (7.7 to 6.8 Ma), but are not simultaneous records of FCCP 1049 [27,28].

Etymology. Dosanko means people and things born in Hokkaido, northern Japan, originated from native horse of Hokkaido.

Diagnosis. Among the Balaenidae, "*Balaenula*" *dosanko* uniquely has a slender zygomatic process, and deep promontorial groove of the pars cochlearis of the periotic (Character 151). "*B.*" *dosanko* can be differentiated from other balaenids except *Morenocetus parvus* by having a posteriorly oriented postorbital process in dorsal view (Character 38). "*B.*" *dosanko* can be differentiated from balaenids except the *Balaena* and *Eubalaena* by having a weakly and laterally projected lateral margin of the nuchal crest.

Comparison with more basal balaenids (*Morenocetus parvus* and *Peripolocetus vexillifer*), "*B.*" *dosanko* can be differentiated by having a more or less the same length of the anterior process and pars cochlearis of the periotic (Character 139), hypertrophied and blade-like lateral tuberosity (Character 144), posterior process of the periotic orienting a right angle to the axis of the anterior process in ventral view (Character 170), laterally reduced involucral ridge in dorsal view (Character 181), dorsolateral surface of involucrum forming a continuous rim (Character 190), and flat anteromedial portion of ventral surface of tympanic bulla (Character 195). Comparison with crown balaenids (*Balaena* spp. and *Eubalaena* spp.), "*B.*" *dosanko* can be differentiated by having a thickened and flat lateral surface of the orbital rim of the supraorbital process (Character 40), pyramidal process (Character 141), laterally exposed and distinct compound posterior process from the lateral skull wall (Character 172), no crest on the parieto-squamosal suture (Character 93), no hypertrophied suprimeatal fossa (Character 162) and no transverse creases on the dorsal surface of the involucrum (Character 191).

4. Description

4.1. Ontogeny

FCCP 1049 possibly be subadult. A caudal vertebra has fused epiphyses. Thoracic vertebrae are not fused with epiphyses. Isolated vertebral epiphyses are preserved. On the skull, the exoccipital/supraoccipital suture is fused, which suggest that FCCP 1049 is older than postnatal [29]. The frontal/parietal, parietal/squamosal, squamosal/exoccipital sutures are fused but visible.

4.2. Skull

General feature of the skull. A preserved left side of the skull has a slender supraorbital process with weakly projected pre and postorbital processes, slender zygomatic process and laterally weakly protoruded nuchal crest.

Premaxilla. A supposed ascending process is preserved. The ascending process is wide and flat (about 87 mm wide, Figs. 2 and 6). Its posterior end still has matrix and does not show the posterior border.

Maxilla. The orbital plate covers anterior border of the supraorbital process. The plate has a transverse ridge on the dorsal surface.

Frontal. The supraorbital process is slender and tilts down laterally. The mid part of the supraorbital process is anteroposteriorly shorter, and the lateral part is anteroposteriorly

longer, because the posterior border of the supraorbital process is anteriorly excavated at the middle (Fig. 2). There is a weak transverse ridge on the dorsal surface of the supraorbital process. The lateral margin of the process in dorsal view is more or less straight. In lateral view (Fig. 5), an anteroposteriorly thin postorbital process projects posteroventrally, and is rounded in dorsoventral view. The preorbital process is more robust than the **post orbital** process. Between the preorbital and postorbital processes form a dorsally excavated orbital region. At the medial end of the frontal contacts with the parietal posteriorly at the anterior portion of the temporal fossa, and with the supraoccipital dorsally as a part of the nuchal crest. The frontal forms a part of the vertex, and is partially covered by the ascending process of the **maxilla**. Medial to the ascending process, the frontal continues to medial, which suggests that the left and right frontals connect each other at the midline. The frontal ridge might be absent, not like *Morenocetus parvus* [2]. In ventrally view, low preorbital and postorbital ridges run from medial to lateral, and its lateral part curve posteriorly.

Parietal. The parietal locates posterior to the frontal and anterior to the squamosal. The parietal forms the anterior part of the temporal fossa (Fig. 5), which is flat. The parietal/squamosal suture is visible. **The alisphenoid is not visible.**

Squamosal. The squamosal has a slender zygomatic process (Fig. 2). The anterior part of the zygomatic process is anteroposteriorly thin, and forms a flat anterior surface for the postorbital process. On the dorsal surface of the zygomatic process, a strong supramastoid crest runs to posteromedially and reaches posterior part of the nuchal crest. The supramastoid crest has a huge anteroposteriorly thin rounded squamosal prominence, dorsal to the postglenoid process and external auditory meatus. The length of the crest is about 170 mm and 65 mm high. On the lateral surface of the base of the zygomatic process shows two square shallow fossae for the sternocephalicus.

In ventral view (Fig. 3), the squamosal has a very faint falciform process, which could be identified using the periotic (Fig. 7). Anterior to the falciform process, there is an anteroposteriorly long shallow groove, which might be a part of foramen ovale. A shallow fossa for the periotic locates medial to the falciform process.

Posterior to the falciform process and lateral to the fossa for the periotic, a large external auditory meatus (23.5 mm long, 26.0+ deep at the lateral end of the meatus) runs transversely. Posterior to the external auditory meatus, there is a huge laterally wider fossa for the posterior process of the periotic (29 mm long, 32 mm high at the lateral end of the fossa). The fossa shows the border between the squamosal and exoccipital. A broken postglenoid process is wide even its anterior and ventral parts are broken away in posterior view (Fig. 4 (a), (b)).

Exoccipital. In dorsal and posterior view (Figs. 2 and 4 (a), (b)), the exoccipital forms a rounded ventrolateral part of the occipital shield. In ventral view, the exoccipital is mediolaterally long plate (about 27 mm long), but the medial end is strongly worn. The lateral part of the exoccipital forms a posterior part of the fossa for the posterior process. The occipital condyle is flat and only weakly project posteriorly.

Supraoccipital. The supraoccipital is a long triangle (Fig. 2). There is a shallow dorsal condyloid fossa dorsal to the occipital condyle. The foramen magnum preserves dorsal side and its dorsal margin becomes thicker.

4.3. Periotic

The periotics (Figs. 7-10 and Table 2) have robust anterior process, small globular pars cochlearis, and large compound posterior process of tympanoperiotic.

The anterior process is short but wide with a prominent lateral tuberosity. The size and shape of the lateral tuberosities and also the posterior processes are slightly different on the right and left sides of FCCP 1049 (Table 2). Anterior to the lateral tuberosity, there is a shallow notch, which might be the anteroexternal sulcus. Medial to the lateral tuberosity, the anterior pedicle of the tympanic bulla is anteroposteriorly long rectangular, and places on the ventral surface of the anterior process. Between the anterior pedicle and lateral tuberosity, there is a large and weakly depressed plane for the sigmoid process of the tympanic bulla (Fig. 8). A shallow wide malleolar fossa is located slightly posterior level of the lateral tuberosity. In **ventral** view, the medial part of the periotic has two large processes: the pyramidal process anteriorly and the dorsal tuberosity posteriorly. Between these processes,

an anteroposteriorly long and strongly curved dorsal rim of the suprameatal fossa [30]. Ventral to the rim, the suprameatal fossa is large and shallow.

An anteroposteriorly long globular pars cochlearis covers the body of the periotic anteriorly as a thin bone and it forms the anterior incisure. In the anterior incisure, there is a huge dorsoventrally long elliptical hiatus fallopii (4.7 mm high, 3.2 mm wide) opening anteriorly. The medial surface of the pars cochlearis has a deep, anteroposteriorly long and weakly curved promontorial groove. Just posterior to the hiatus fallopii, a large internal acoustic meatus (9.1 mm long, 5.2 mm high) opens medially and contains the proximal opening of the facial canal and dorsal vestibular area. The proximal opening of the facial canal opens (5.0 mm high, 4.3 mm long). Just posterior to the facial canal, and clearly separated from it by a low transverse crest, there is a circular dorsal vestibular area (5.2 mm high, 5.9 mm long). In the dorsal vestibular area, there is a small foramen singulare (1.0 mm diameter) anterodorsally and two depressed areas, which might be the spiral cribriform tract and area cribrosa media posteroventrally. The spiral cribriform tract and area cribrosa media are connected and form a weakly curved depression. Posterior to these openings, there are two foramina: a larger vestibular aqueduct (4.0 mm high, 4.8 mm long), and a smaller aperture for the cochlear aqueduct (2.5 mm high, 1.2 mm long). The posterior end of the pars cochlearis has a large elliptical fenestra rotunda (4.4 mm long, 8.1 mm wide), and the anterior border of the fenestra rotunda runs to the medial end of the pars cochlearis and projects as a anteroposteriorly short plate. The posterior end of the pars cochlearis is formed by the caudal tympanic process, which has transversely long sharp edge.

A large fenestra ovalis (6.0 mm long, 4.3 mm wide) is located posterior to the malleolar fossa, lateral to the pars cochlearis (Fig. 10, (i)). Posterior to the fenestra ovalis, there is a deep and long stapedial muscle fossa (15.5 mm long, 4.5 mm wide). Medial to the fenestra ovalis, a long fossa for the anterior part of the facial sulcus. The facial sulcus runs posteriorly, and forms a deep groove on the posterior surface of the compound posterior process of the tympanoperiotic. Lateral to the facial sulcus, an anteroposteriorly long elliptical posterior pedicle of the tympanic bulla shows broken surface. The posterior process becomes dorsoventrally high at the middle of the process, and the lateral end is dorsoventrally thin.

4.4. Tympanic bulla

The tympanic bullae (Figs. 7, 8, 11, 12 and Table 2) are long and wide with wide anterior portion in ventral view.

The anterior **robe** of the tympanic bulla is swollen laterally. On the ventral to lateral surface of the anterior **robe** has a blunt ridge, which is a part of the main ridge running to the medial surface. Posterior to the anterior **robe**, a large triangular sigmoid process projects laterally. Its dorsal most point bends posteriorly (Fig. 12 (a)). The sigmoid process has thick margins, about 8.0 mm (anteroposterior length). Anterior to the sigmoid process, a deep lateral furrow runs transversely and separates the sigmoid process and anterior **robe**. Anterior to the sigmoid process, the rim of the outer lip becomes thicker, which is the malleolar ridge. The malleolar ridge contains structures such as the sulcus for the chorda tympani and fossa for the malleus dorsally. An anteroposteriorly long sulcus for the chorda tympani locates anterior to a long fossa for the malleus. The lateral surface of the malleolar ridge has several oblique striae. The involucrem is posteriorly wider (Fig. 12, (g)). Its lateral surface has several transverse grooves. A broken base of the posterior pedicle locates at the posterior end of the involucrem on the lateral surface. The broken section of the posterior pedicle is anteroposteriorly long elliptical (16.6 mm long, 9.0 mm wide). Medial to the posterior pedicle, there is a smooth conical process (Fig. 12, (g)). On the medial surface of the tympanic bulla, there are stronger main ridge and weaker involucrem ridge running anteroposteriorly. Between them, there is a median furrow, which is deep anteriorly but weak posteriorly. The main ridge runs from the anterior lobe to the posterior end of the tympanic bulla, forming the anterior most point and also the medial margin of the tympanic bulla.

4.5. Mandible

An incomplete right mandible is weakly laterally bowed, especially anterior part (Figure 13, Table 1). The most anterior part has an anteroposteriorly long mental foramen, which also

opens anteriorly. The anterior part of the mental foramen is shallower and wider. Medially, there is a flat area for mandibular symphysis, which is restricted by a wide symphyseal groove running anteroposteriorly. The symphyseal groove is anteriorly wider and shallower, and its posterior end runs on the medial surface of the mandible. The posterior part preserves a broken coronoid process. The mandibular foramen (about 50 mm high and 40 mm wide at the broken posterior margin) locates dorsally in medial view.

4.6. Postcranial skeleton

Vertebra. Two thoracic vertebrae (Fig. 14 (a) and (b)) show dorsally wider triangular body with a ventral keel, large neural canal and long transverse process. A fossa for the rib locates ventral to the transverse process. The vertebral epiphyses are all removed. A **caudal vertebra** (Fig. 14 (c)) shows the body with fused epiphyses. Supposed anterior articular surface is slightly larger than posterior one (anterior surface: 86 mm wide and 93 mm high, posterior surface: 80 mm wide and 90 mm high, length of the body is 71 mm). Seven isolated epiphysis (Fig. 14 (d) to (j)) are dorsally wider triangular to laterally wider elliptical. The smallest epiphysis number D is 90 mm wide and 81 mm high. The largest epiphysis number J is 131 mm wide and 112 mm high

Presternum. Preserved part of the presternum shows that symmetrical anterior part is broken and dorsoventral surfaces are worn (Fig. 14, (k), Table 3). A side is convex, which can be identified as the ventral side (Fig. 14, (k)). On the lateral margin, there are a couple of thicker part, which might be a surface for the ribs.

Rib. Five ribs are preserved (Fig. 15). The rib number (b) in Figure 15 is wide, which might be the most anterior rib. The longest one, number E is about 560 mm long and 65 mm wide, and its restoration using clay might overestimate its length.

5. Discussion

5.1. Phylogenetic analysis

The phylogenetic position of “*Balaenula*” *dosanko* sp. nov. (FCCP 1049) was analyzed using the matrix of Buono et al. [2], which was derived from the matrix of Marx and Fordyce [31]. This study modified codings of FCCP 1049 with direct examination (Supplementary 1 to 4) and contains 257 morphological characters and 43 taxa. Percentages of coded data of FCCP 1049 are 46 % (includes soft tissue characters), 48 % (excludes soft tissue) and 89% for the ear bones.

The matrix was managed using Mesquite 2.75 [32]. Analysis was performed with TNT version 1.5 [33]. All of the characters were treated as unweighted and unordered with backbone constraint of extant taxa, based on a topology of the molecular tree [34]. The analysis used New Technology Search with recover minimum length trees = 1000 times.

The phylogenetic analysis shows two shortest trees of 947 steps each. The strict consensus trees (Fig. 16 and Supplementary 5) are different from that of the equally weighted analysis of Buono et al. [2] in the most basal balaenid is *Morenocetus parvus* but *Peripolocetus vexillifer*, and “*B.*” *dosanko* + *Balaenula astensis* + *Balaenella brachyrhynchus* + crown clade are appeared as an unsolved polytomy. Indeed, the branch lengths of the previous phylogenetic hypothesis around them were relatively low [2].

5.2. Comparison with *Balaenula* and *Balaenella*

“*Balaenula*” *dosanko* is identical from the all named balaenids, but hard to determine the genus belonged. Because the results of this phylogenetic analysis show “*Balaenula*” *dosanko* + *Balaenula astensis* + *Balaenella brachyrhynchus* + crown clade as an unsolved polytomy. Here, we compare “*B.*” *dosanko* with the *Balaenula* then *Balaenella* (Table 4).

Several diagnoses of the genus *Balaenula* was reported by Bisconti [7]. A diagnosis for the genus, having a not protruding nuchal crest is not seen on FCCP 1049. FCCP 1049 shows a lateral expansion at the posterior part of the supraoccipital crest). One comparable diagnosis of the genus, such as having a low and rounded supramastoid crest (the lateral squamosal crest in the study) is difficult to consider. Because some non *Balaenula* species such as *Eubalaena belgica* also shows low and rounded supramastoid crest (see later). Thus, not only the results of phylogenetic analyses, but also the previously mentioned diagnoses for the *Balaenula* support that “*B.*” *dosanko* might not belong to the *Balaenula*. However,

here we present “*Balaenula*” *dosanko* in the genus, preliminary.

The type species of the genus *Balaenula* is doubtful to recognize as an individual [2]. Indeed, the long and triangular anterior process of the periotic on the type specimen is not likely belongs to the Balaenidae, but Balaenopteroidea. Based on illustrations of previous studies [7,35], the periotic seems having some balaenopteriid diagnoses such as a narrow and triangular anterior process [36,37], but lack other diagnoses for the family such as a well defined fossa for the malleus [37] and transversely elongated pars cochlearis [38]. Thus, it might be a stem balaenopteroid. Its overall shape in ventral view is similar to that of *Tiphyocetus temblorensis* of Kellogg [39] by having a slender anterior process and globular pars cochlearis, but morphologies of the internal acoustic meatus and lateral tuberosities are different.

Here, comparisons on “*B.*” *dosanko* with closely related genus *Balaenula* and *Balaenella* (see Table 4). Comparison with the genus *Balaenula*, “*B.*” *dosanko* shows weaker pre and postorbital processes in lateral view, and weaker orbitotemporal crest only on the medial part of the supraorbital process. Comparison with *Balaenula balaenopsis*, “*B.*” *dosanko* shows more or less straight lateral margin of the supraorbital process, and much weaker supramastoid crest. Comparison with *Balaenula astensis*, FCCP 1049 shows a more slender supraorbital process, no tubercle at the junction of the parieto-squamosal suture and supraoccipital (Character 82), lateral projection of the nuchal crest, and more posteriorly located posterior supraoccipital fossa.

Comparison with *Balaenella brachyrhynchus*, “*B.*” *dosanko* shows a more strongly curved posterior margin of the supraorbital process in dorsal view, much weaker supramastoid crest, more rounded postorbital process in dorsal view, ventrally almost closed frontal groove by the pre and postorbital ridges, much smaller lateral process of the periotic, thinner medial margin of the fenestra rotunda, much weaker outer posterior prominence of the tympanic bulla, much stronger inner posterior prominence, and more rounded anteromedial angle of the tympanic bulla in ventral and dorsal views. Of note, *Balaenella brachyrhynchus* does not preserve the anterior tip of the zygomatic process. Some *Balaenula* sp. specimens from the late Miocene to early Pliocene, and Late Pliocene of California have mentioned, but not been photographed and illustrated [40].

5.3. The supramastoid crest

The supramastoid crest of balaenids are generally developed. But, morphological change of the supramastoid crest can not be thought simple, like developing from incipient to large through balaenid evolution. The oldest known balaenid *Morenocetus parvus* shows small supramastoid crest on Holotype, but just a ridge along the dorsal surface of the zygomatic process on referred specimen (MPL 5-15) [2].

The modern balaenids show varied condition of the supramastoid crest. *Balaena mysticetus* shows both small (a juvenile individual described by Nishiwaki and Kasuya (1970)) and well developed (USNM 257513, see figure 5 of Bisconti [7]) conditions, as Field et al. [42] mentioned. *Eubalaena australis* shows small ones (a young individual in Best [43] and USNM 26712, see figure 4 of Bisconti [7]). Possibly the development of the supramastoid crest depends on growing. Some fossil species belonging to the genus *Balaena* and *Eubalaena* such as *Eubalaena belgica* and *E. ianatrix* also have small ridges [3,44], but *B. montalionis* and *Eubalaena* sp. (IMNH 9598) have large ridges [11].

As mentioned above, having a low and rounded supramastoid crest was considered as a diagnosis of the genus *Balaenula*. However, early Pliocene balaenids (“*B.*” *dosanko*, *Balaenula* spp. and *Balaenella brachyrhynchus*) have relatively large supramastoid crests among the Balaenidae. Among them, the conditions of the supramastoid crest are varied as compared above (see also Table 4), especially *Balaenula balaenopsis* has larger and dorsally strongly projected supramastoid crest than those of “*B.*” *dosanko*, *Balaenula astensis* and *Balaenella brachyrhynchus*.

The rounded and low supramastoid crest emerged among the Balaenidae already during the early Miocene as incipient condition, then enlarged by the early Pliocene, and some kept the condition or others got the crest smaller secondary. In short, the morphological change of the supramastoid crest is not simple. The supramastoid crest is attached to the temporal fascia [45]. Thus, these conditions of the supramastoid crest might be related to size

and/or orientation and the temporal muscle, but is also related to many factors (width of the supraorbital process, degree of telescoping, orientation of the skull, arching of the rostrum, width, height and length of the temporal fossa).

6. Conclusion

“*Balaenula*” *dosanko* sp. nov. (FCCP 1049) represents an archaic balaenid from the early Pliocene, Zanclean (3.5 to 5.2 Ma) lower part of the Chippubetsu Formation, Hokkaido, northern Japan. The fossil balaenid is distinguishable from another balaenids by having a slender zygomatic process, deep promontorial groove of the pars cochlearis of the periotic, and a weak orbitotemporal crest only on the medial part of the supraorbital process. “*B.*” *dosanko* does not show previously introduced diagnoses for the genus *Balaenula*. The result of phylogenetic analysis places “*B.*” *dosanko* among an unsolved polytomy of *Balaenula astensis* + *Balaenella brachyrhynchus* + crown clade. “*B.*” *dosanko* is different from these closely related taxa in many points. Thus, “*B.*” *dosanko* is preliminarily belonged to the genus “*Balaenula*”, in this study. The genus *Balaenula* had been treated as “a taxonomical basket, where all the small-sized balaenids were put in. In addition, the type species of the genus is doubtful to recognize as an individual. It might be better to wait taxonomical works by adding more related species in the future.

Data accessibility. The data for the phylogenetic analysis is available as electronic supplementaries.

Authors’ contributions. Y. T. carried out the phylogenetic analysis and write the paper. H. F. and M. K. revised the manuscript.

Competing interests. There are no competing interests.

Funding. There are no fundings.

Acknowledgements. We thank to J. Takahashi, N. Mita and others to discover, and 100 
[revised manuscript text omitted]

Figure 2. Skull of FCCP 1049, “*Balaenula*” *dosanko* in dorsal view. (a) photo; (b) line art.

Figure 3. Skull of FCCP 1049, “*Balaenula*” *dosanko* in ventral view. (a) photo; (b) line art.

Figure 4. Skull of FCCP 1049, “*Balaenula*” *dosanko* in anterior and posterior views. (a) photo; (b) line art; (c) photo; (d) line art.

Figure 5. Skull of FCCP 1049, “*Balaenula*” *dosanko* in lateral view. (a) photo without the supraorbital process in left lateral view; (b) photo with the supraorbital process in left lateral view; (c) photo in right lateral view; (d) line art of (b).

Figure 6. Skull, vertex of FCCP 1049, “*Balaenula*” *dosanko* in dorsal view. (a) photo; (b) line art.

Figure 7. Skull and left ear bones of FCCP 1049, “*Balaenula*” *dosanko* in ventral view. (a) with periotic and without tympanic bulla; (b) with periotic and tympanic bulla.

Figure 8. Right periotic and tympanic bulla connections of FCCP 1049, “*Balaenula*” *dosanko* in ventrolateral view.

Figure 9. Periotics of FCCP 1049, “*Balaenula*” *dosanko*; (a)-(f) left periotic; (g)-(i) right periotic; (a) and (h) ventral view; (b) and (g) medial view; (c) dorsal view; (d) lateral view; (e) posterior view; (f) anterior view; (i) posteroventral view.

Figure 10. Key features of periotics of FCCP 1049, “*Balaenula*” *dosanko*; (a)-(f) left periotic; (g)-(i) right periotic; (a) and (h) ventral view; (b) and (g) medial view; (c) dorsal view; (d) lateral view; (e) posterior view; (f) anterior view; (i) posteroventral view.

Figure 11. Tympanic bullae of FCCP 1049, “*Balaenula*” *dosanko*; (a)-(f) right tympanic

bulla; (a) lateral view; (b) ventral view; (c) anterior view; (d) medial view; (e) posterior
view; (f) dorsal view; (g) left tympanic bulla in lateral view.

**Figure 12.** Key features of tympanic bullae of FCCP 1049; (a)-(f) right tympanic bulla; (a)
lateral view; (b) ventral view; (c) anterior view; (d) medial view; (e) posterior view; (f)
dorsal view; (g) left tympanic bulla in lateral view.

**Figure 13.** Right mandible of FCCP 1049, "*Balaenula*" *dosanko*; (a) lateral view; (b) dorsal
view; (c) medial view; (d) to (f) cross sections; (g) medial view of posterior end.

**Figure 14.** Postcranial skeleton of FCCP 1049, "*Balaenula*" *dosanko*; (a) and (b) thoracic
vertebrae; (c) caudal vertebra; (d) to (j) isolated vertebral epiphyses; (k) presternum in
ventral view.

**Figure 15.** Ribs of FCCP 1049, "*Balaenula*" *dosanko* (a) to (c) right ribs; (d) and (f) left
ribs.

**Figure 16.** Strict consensus tree of two most parsimonious trees to show the phylogenetic
position of "*Balaenula*" *dosanko* sp. nov. The most parsimonious trees were 974 steps. The
clades Aetiocetidae, Eomysticetidae, Balaenidae, Cetotheriidae and a clade comprising
*Isanacetus*, *Parietobalaena* and related taxa are collapsed. (See Supplemental material file 5
for cladogram with all taxa shown.) The numbers on the each branch are branch lengths. The
numbers next to the branch lengths show the percentages of the shortest trees supporting
each node. 100% supported nodes are omitted the percentages.

**Table 1.** Measurements in cm of "*Balaenula*" *dosanko* (FCCP 1049) skull and mandible
following Buono *et al.* [2], and Tanaka and Taruno [48]. For skull and mandible, distances
are either horizontal or vertical, unless identified as point to point. + shows an incomplete
measurement, because of erosion. * shows measurements taken from only one side.

**Table 2.** Measurements in mm of "*Balaenula*" *dosanko* (FCCP 1049) periotic and tympanic
bulla. Dimensions follow [49]. Distances are either horizontal or vertical, unless identified as
point to point. + shows an incomplete measurement, because of erosion. Measurements of
*Balaenulla balaenopsis* CtM 858a was taken from table 7 of Bisconti [7].

**Table 3.** Measurements in mm of "*Balaenula*" *dosanko* (FCCP 1049) vertebrae and
presternum. Dimensions follow Tanaka and Taruno (submitted). Measurements are rounded
to the nearest 0.5 cm. Distances are either horizontal or vertical, unless identified as point to
point. + shows an incomplete measurement, because of erosion.

**Table 4.** Comparison between "*Balaenula*" *dosanko* (FCCP 1049) and related fossil
balaenids using FCCP 1049 specimen and previous publications [4,35].

**Supplementary 1.** Datamatrix in nex format.

**Supplementary 2.** Datamatrix in tnt format.

**Supplementary 3.** Character list.

**Supplementary 4.** Modified codings from Buono et al (2017).

**Supplementary 5.** Full figure of the analysis

**Supplementary 6.** Tree file

Figure 1. Maps showing the locality of FCCP 1049, "Balaenula" dosanko The base maps A and B are modified from Tanaka and Kohno [46], and Wada and Akiyama [47].

183x90mm (300 x 300 DPI)

Figure 2. Skull of FCCP 1049, "Balaenula" dosanko in dorsal view. (a) photo; (b) line art.

189x237mm (300 x 300 DPI)

Figure 3. Skull of FCCP 1049, "Balaenula" dosanko in ventral view. (a) photo; (b) line art.

187x241mm (300 x 300 DPI)

Figure 4. Skull of FCCP 1049, "Balaenula" dosanko in anterior and posterior views. (a) photo; (b) line art; (c) photo; (d) line art.

188x163mm (300 x 300 DPI)

Figure 5. Skull of FCCP 1049, "Balaenula" dosanko in lateral view. (a) photo without the supraorbital process
in left lateral view; (b) photo with the supraorbital process in left lateral view; (c) photo in right lateral view;
(d) line art of (b).

189x161mm (300 x 300 DPI)

Figure 6. Skull, vertex of FCCP 1049, "Balaenula" dosanko in dorsal view. (a) photo; (b) line art.

182x60mm (300 x 300 DPI)

Figure 7. Skull and left ear bones of FCCP 1049, "Balaenula" dosanko in ventral view. (a) with periotic and without tympanic bulla; (b) with periotic and tympanic bulla.

94x88mm (300 x 300 DPI)

Figure 8. Right periotic and tympanic bulla connections of FCCP 1049, "Balaenula" dosanko in ventrolateral
view.

89x115mm (300 x 300 DPI)

Figure 9. Periotics of FCCP 1049, "Balaenula" dosanko; (a)-(f) left periotic; (g)-(i) right periotic; (a) and (h)
ventral view; (b) and (g) medial view; (c) dorsal view; (d) lateral view; (e) posterior view; (f) anterior view;
(i) posteroventral view.

185x236mm (300 x 300 DPI)

Figure 10. Key features of periotics of FCCP 1049, "Balaenula" dosanko; (a)-(f) left periotic; (g)-(i) right periotic; (a) and (h) ventral view; (b) and (g) medial view; (c) dorsal view; (d) lateral view; (e) posterior view; (f) anterior view; (i) posteroventral view.

187x236mm (300 x 300 DPI)

Figure 11. Tympanic bullae of FCCP 1049, "Balaenula" dosanko; (a)-(f) right tympanic bulla; (a) lateral view; (b) ventral view; (c) anterior view; (d) medial view; (e) posterior view; (f) dorsal view; (g) left tympanic bulla in lateral view.

187x235mm (300 x 300 DPI)

Figure 12. Key features of tympanic bullae of FCCP 1049; (a)-(f) right tympanic bulla; (a) lateral view; (b) ventral view; (c) anterior view; (d) medial view; (e) posterior view; (f) dorsal view; (g) left tympanic bulla in lateral view.

187x236mm (300 x 300 DPI)

Figure 13. Right mandible of FCCP 1049, "Balaenula" dosanko; (a) lateral view; (b) dorsal view; (c) medial view; (d) to (f) cross sections; (g) medial view of posterior end.

184x99mm (300 x 300 DPI)

Figure 14. Postcranial skeleton of FCCP 1049, "Balaenula" dosanko; (a) and (b) thoracic vertebrae; (c) caudal vertebra; (d) to (j) isolated vertebral epiphyses; (k) presternum in ventral view.

187x61mm (300 x 300 DPI)

Figure 15. Ribs of FCCP 1049, "Balaenula" dosanko (a) to (c) right ribs; (d) and (e) left ribs.

63x115mm (300 x 300 DPI)

Figure 16. Strict consensus tree of two most parsimonious trees to show the phylogenetic position of "Balaenula" dosanko sp. nov. The most parsimonious trees were 974 steps. The clades Aetiocetidae, Eomysticetidae, Balaenidae, Cetotheriidae and a clade comprising Isanacetus, Parietobalaena and related taxa are collapsed. (See Supplemental material file 5 for cladogram with all taxa shown.) The numbers on the each branch are branch lengths. The numbers next to the branch lengths show the percentages of the shortest trees supporting each node. 100% supported nodes are omitted the percentages.

174x54mm (300 x 300 DPI)

Skull	
Bizygomatic width	59.0*
Width of the skull at the level of the exoccipitals	34.0*
Supraoccipital width anterior to foramen magnum	32.2*
Supraoccipital width at mid-length	18.0*
Supraoccipital width at 10 cm to the anterior margin	12.0*
Supraoccipital length	38.8
Transverse diameter of left occipital condyle	7.8
Dorsoventral diameter of left occipital condyle	9.4+
Transverse diameter of foramen magnum	7.2
Distance between lateral border of left occipital condyle	22.0
Width of the occipital condyles plus foramen magnum	18.8 plus
Anteroposterior diameter of the supraorbital process	16.0
Transverse diameter of the supraorbital process	48.2
Anteroposterior diameter of the supraorbital process	10.2
Width of the optic canal in its medial portion	0.7
Width of the optic canal in its lateral portion	11.2
Length of the optic canal	40.0+
Orbital length (between preorbital and postorbital foramina)	13.0
Anteroposterior diameter of the temporal fossa	27.5
Transverse diameter of the temporal fossa	39.5
Mandible	
Length of mandible, as preserved in straight line	180.0
Height of mandible, from coronoid process to ventral margin	15.5+
Maximum preserved height of mandible	15.5+
Maximum preserved width of mandible	74.0
Height of body at just anterior to coronoid process	15.3
Width of body at just anterior to coronoid process	74.0

			Balaenula balaeno
Periotic	left	right	
Length of anterior process of periotic	34.7	36.0	32.0
Width of anterior process of periotic	46.4	60.0	43.5
Length of anterior process, from anteroventral angle to anterior incisure	24.3	32.7	-
Maximum transverse width of the body of periotic	46.4	60.0	-
Length of pars cochlearis	30.2	31.4	54.0
Width of pars cochlearis	16.8	17.7	30.0
Dorsoventral diameter of internal acoustic meatus			
	9.3	9.3	4.3
Length of internal acoustic meatus	13.7	13.9	6.8
Greatest length of aperture for the cochlear aqueduct	1.2	-	-
Greatest length of aperture for the vestibular aqueduct	4.8	5.0	-
Dorsoventral diameter of fenestra rotunda			
	4.1	-	3.4
Width of fenestra rotunda	7.5	-	4.3
Length of fenestra ovalis	6.0	-	3.7
Width of fenestra ovalis	4.3	-	5.4
Greatest length of malleolar fossa	7.0	5.6	-
Greatest length of ventral opening of the facial canal tuberosity	-	26.4	-
	42.7	54.4	-
Length of the posterior process of the periotic	139.6	114.6	-
Bulla			
Greatest length, in lateral view	94.7	94.4	-
Greatest length of tympanic cavity	-	72.8	-
Greatest width, in lateral view	-	74.2	-
Greatest height, from tip of sigmoid process to the ventral-most point in posterior view	-	65.2	-
Height of involucrum at the anteriormost point of the posterior pedicle	30.8	34.0	-
Length of anterior lobe, from lateral furrow to anterior tip of tympanic bulla	-	42.0	-
Greatest length of posterior pedicle at base	16.6	16.9	-

Thoracic vertebra A in Fig. 13

maximum preserved length	77.0
maximum preserved height	270.0
maximum preserved width	191.0+
length of neural spine	48.0
height of neural spine	115.0
length of body	52.0
width of body anteriorly	122.0
height of body anteriorly	92.0
width of body posteriorly	144.0
height of body posteriorly	90.0

Thoracic vertebra B in Fig. 13

maximum preserved length	86.0
maximum preserved height	172.0
maximum preserved width	237.0
length of body	49.0
width of body anteriorly	125.0
height of body anteriorly	77.0
width of body posteriorly	132.0
height of body posteriorly	92.0

Sternum

maximum preserved length	118.0
maximum preserved height	18.0
maximum preserved width	83.0

Sheet1

	"Balaenula" dosanko	Balaenula balaenopsis	Balaenula astensis	Balaenella brachyrhynus
pre and postorbital processes	weak	strongly project	strongly project	weak
lateral margin of supraorbital process in supraorbital process	straight	strongly excavated	straight	straight
orbitotemporal crest	slender weak, only on medial part of supraorbital	slender strong	robust strong	slender weak, only on medial part of supraorbital process
supramastoid crest	weaker	stronger	weaker	weaker
zygomatic process	slender	robust	robust	-
Distinct tubercle at jugular process	absent	-	absent	present
lateral margin of nuchal crest	laterally projected	-	straight	laterally projected
fossa	absent	-	present	absent
posterior supraoccipital fossa position	just anterior to occipital condyles	-	more anterior	-

Appendix B

Modified codings from Buono et al. (2017)

This study modified codings on "*Balaenula*" *dosanko* (FCCP 1049) with direct examination. "*Balaenula*" *dosanko* was coded as "*Balaenula* sp." in Buono et al. (2017).

Skull

13. Distinct pocket between the ascending process of the maxilla dorsally and the supraorbital process ventrally: absent (0); present (1).

"*Balaenula*" *dosanko* 0 to ?

27. Teeth in adult individuals: present (0); absent or vestigial (1).

"*Balaenula*" *dosanko* 1 to ?

31. Anterior edge of supraorbital process lateral to ascending process of maxilla with the skull in dorsal view: oriented transversely or pointing anteriorly (0); pointing posteriorly (1); linguiform and tapering to a point (2).

"*Balaenula*" *dosanko* 0 to ?

33. Transverse width of anterior edge of supraorbital process lateral to ascending process of maxilla: longer than or equal to the combined transverse width of the adjacent rostral bones, as measured from the sagittal plane to the lateral border of the ascending process of the maxilla (0); shorter than the combined transverse width of the adjacent rostral bones (1).

"*Balaenula*" *dosanko* 0 to ?

35. Supraorbital process of frontal in anterior view: horizontal or nearly horizontal (0); gradually slopes away lateroventrally from the skull vertex (1); as state 1 but with the lateral portion of the supraorbital being nearly horizontal, thus causing the latter to appear concave in anterior view (2); abruptly depressed to a level noticeably below the vertex, with the lateral skull wall above the supraorbital formed by both parietal and frontal (3).

"*Balaenula*" *dosanko* 0 to ?

Commented [M1]: Why this character is coded as ? If the anterior margin of the supraorbital process is preserved?

Commented [M2]: Why this character is coded as ? If the supraorbital process is preserved?

38. Postorbital process in dorsal view: oriented posteriorly (0); oriented laterally (1); oriented posterolaterally (2); short and not markedly projecting in any direction (3).

"Balaenula" dosanko 1 to 0

Balaenula astensis 1 to 0

40. Orbital rim of supraorbital process of frontal in lateral view: dorsoventrally thin (0); thickened with a flat lateral surface (1); thickened with a rounded lateral surface (2).

"Balaenula" dosanko 0 to 1

Balaenula astensis 0 to 1

Commented [M3]: The lateral surface of the orbital rim is not flat in both taxa (see for example *Megapera* which exhibits this condition). I suggest to code both as state 2

72. Intertemporal constriction: longer anteroposteriorly than wide transversely (0); as state 1, but with the temporal fossa forming a large parasagittal oval (1); wider transversely than long anteroposteriorly (2).

"Balaenula" dosanko 2 to ?

Commented [M4]: Half of the intertemporal constriction is preserved, so I think that is possible to codified this character as state 2

73. Exposure of frontal on skull vertex: broadly exposed (0); anteroposteriorly compressed or absent (1).

"Balaenula" dosanko - to 1

75. Outline of fronto-parietal suture: straight or lobate (0); frontals projects posteriorly along the sagittal plane and separate the left and right parietal anteriorly (1); highly irregular (2).

"Balaenula" dosanko - to 1

85. Zygomatic process of squamosal dorsoventrally expanded in lateral view: absent (0); present, with the zygomatic process being distinctly higher dorsoventrally than wide transversely (1).

"Balaenula" dosanko 1 to ?

90. Apex of zygomatic process of squamosal deflected anteroventrally: absent (o); present (i).

Commented [M5]: Why is coded as ?, the left zygomatic process is preserved...

"Balaenula" dosanko 1 to ?

Commented [M6]: Same comment above

91. Supramastoid crest of zygomatic process of squamosal (skull in lateral view): present (o); absent (i).

"Balaenula" dosanko 1 to o

92. Size of squamosal including zygomatic and postglenoid processes: longer anteroposteriorly than high dorsoventrally, or about as high as long (o); distinctly higher than long (i).

"Balaenula" dosanko 1 to ?

Commented [M7]: Perhaps the postglenoid process is not preserved, it is clear the squamosal could be codified as state 1

Periotic

134. Anterior process of periotic in lateral view: squared off or rounded (o); triangular (i); anterior border of process is two-bladed and L-shaped (2).

"Balaenula" dosanko 2 to o

Balaenula astensis ? to o

135. Shape of anteroventral angle of anterior process of periotic in medial or lateral view: rounded or forms a relatively blunt angle (o); slender and tapering to a point (i).

Balaenula astensis ? to o

137. Dorsal deflection of anterodorsal corner of anterior process: absent (o); present (i).

"Balaenula" dosanko 1 to ?

151. Promontorial groove on medial side of pars cochlearis: present, but relatively shallow (0); present and deeply excavated (1); present and forming a distinct constriction, separating a smooth and rounded ventral portion of the pars cochlearis from a flattened and striated dorsal one (2); absent (3).

"*Balaenula*" dosanko 0 to 1

165. Size of proximal opening of facial canal: no more than half the size of the internal acoustic meatus (0); more than half the size of the internal acoustic meatus (1).

"*Balaenula*" dosanko 0 to 1

Commented [M8]: Please check, it is look like that the size of the proximal opening of facial canal is not more than half the size of the internal acoustic meatus

170. Orientation of compound posterior process in ventral view, with periotic being in situ: oriented posterolaterally with respect to the longitudinal axis of the anterior process of the periotic (0); oriented at a right angle to the axis of the anterior process (1).

"*Balaenula*" dosanko 1 to 0

Tympanic bulla

175. Anterior portion of bulla transversely wider than posterior portion in ventral view: absent (0); present (1).

"*Balaenula*" dosanko 1 to ?

Commented [M9]: This character can also be codified.

176. In situ orientation of main axes of tympanic bullae in ventral view: diverging posteriorly (0); parallel (1); diverging anteriorly (2).

"*Balaenula*" dosanko - to 2

187. Medial lobe of tympanic bulla (*sensu* Boessenecker & Fordyce, 2015): present as distinct lobe and transversely wider than its lateral counterpart (0); present but subequal in width to the lateral lobe or smaller (1); absent or indistinct (2).

"*Balaenula*" dosanko 1 to 2

Commented [M10]: Are you sure medial lobe is indistinct?

Sternum

233. Sternum: composed of several bones (o); composed of one bone (i).

"Balaenula" dosanko 1 to ?

Appendix C**ROYAL SOCIETY
OPEN SCIENCE****A new member of fossil balaenid (Mysticeti, Cetacea) from
the early Pliocene of Hokkaido, Japan**

Journal:	Royal Society Open Science
Manuscript ID	RSOS-192182
Article Type:	Research
Date Submitted by the Author:	18-Dec-2019
Complete List of Authors:	Tanaka, Yoshimasa; Osaka Museum of Natural History; Hokkaido University Museum, Division of Academic Resources and Specimens; Numata Fossil Museum Furusawa, Hitoshi; Sapporo Museum Activity Center Kimura, Masaichi; Hokkaido University of Education
Subject:	Palaeontology < EARTH SCIENCES
Keywords:	Balaenidae, New species, Zanclean, Balaenula balaenopsis, supramastoid crest
Subject Category:	Earth science

**Author-supplied statements**

Relevant information will appear here if provided.

**Ethics**

*Does your article include research that required ethical approval or permits?:*

This article does not present research with ethical considerations

*Statement (if applicable):*

CUST_IF_YES_ETHICS :No data available.

**Data**

*It is a condition of publication that data, code and materials supporting your paper are made publicly*
*available. Does your paper present new data?:*

Yes

*Statement (if applicable):*

The data for the phylogenetic analysis is available as electric supplementaries.

**Conflict of interest**

I/We declare we have no competing interests

*Statement (if applicable):*

CUST_STATE_CONFLICT :No data available.

**Authors' contributions**

This paper has multiple authors and our individual contributions were as below

*Statement (if applicable):*

Y. T. carried out the phylogenetic analysis and write the paper. H. F. and M. K. revised the
manuscript.

<Title page>

1) A new member of fossil balaenid (Mysticeti, Cetacea) from the early Pliocene of
Hokkaido, Japan

2) Yoshihiro Tanaka ^{1,2,3} Hitoshi Furusawa⁴ Masaichi Kimura ^{3,5}

¹ Osaka Museum of Natural History, Nagai Park 1-23, Higashi-Sumiyoshi-ku, Osaka, 546-
0034, Japan

tanaka@mus-nh.city.osaka.jp

² Division of Academic Resources and Specimens, Hokkaido University Museum

³ Numata Fossil Museum

⁴ Sapporo Museum Activity Center

hitoshi.furusawa@city.sapporo.jp

⁵ Hokkaido University of Education

mkimura1313@yahoo.co.jp

3) Corresponding author: Yoshihiro TANAKA

Osaka Museum of Natural History

Nagai Park 1-23, Higashi-Sumiyoshi-ku, Osaka, 546-0034, JAPAN

Tel: +81-(0)6-6697-6222

E-mail: tanaka@mus-nh.city.osaka.jp

Abstract

The family Balaenidae includes two genus and four extant species, and at least eight extinct species. The oldest known record of the members of the Balaenidae is known from the early Miocene, but still need more early members of the family to provide better phylogenetic hypotheses. FCCP 1049 from lower part of the Chippubetsu Formation, Fukagawa Group (3.5 to 5.2 Ma, Zanclean, early Pliocene) was preliminary described and identified as *Balaenula* sp. by Furusawa and Kimura in 1982. Later works discussed that FCCP 1049 is different from the genus, and is placed in different clade from *Balaenula astensis*. In this study, FCCP 1049 is re-described and named as "*Balaenula*" *dosanko* sp. nov. FCCP 1049 is distinguishable from another balaenids by having a slender zygomatic process, and deep promontorial groove of the pars cochlearis of the periotic. The result of phylogenetic analysis places "*Balaenula*" *dosanko* among an unsolved polytomy of *Balaenula astensis* + *Balaenella brachyrhynchus* + crown clade. "*Balaenula*" *dosanko* is preliminary belonged to the genus "*Balaenula*", in this study.

ADDITIONAL KEYWORDS: Balaenidae - new species - Zanclean - *Balaenula balaenopsis* – supramastoid crest -

1. Introduction

A baleen whale group, the family Balaenidae includes two genera and four extant species, and at least one extinct species. The modern balaenid body length reach 17 to 20 m [1]. The oldest known nominal species of the members of the Balaenidae is known from the early Miocene [2], and its body length was estimated as 4.8 to 6.2 m [3]. Our knowledge of balaenid diversity and gigantism are growing [2–6], but still need more early members of the family to hypothesize their phylogeny.

A Pliocene genus *Balaenula* had been treated as “a taxonomical basket, where all the small-sized balaenids were put” in history [7]. The first species of the genus *Balaenula balaenopsis* from Antwerp was established by Van Beneden [8]. But, the holotype is doubtful to recognize as an individual [2]. Supposed similar case of establishing new species with mixed individuals was happened on a cetotheriid, *Herpetocetus scaldiensis* of Van Beneden [8], which was summarized by Deméré et al. [9]. The second species, *Balaenula astensis* from the late early Pliocene of Villafranca d’Asti was established by Trevisan [10] and re-described by Bisconti [11].

Furusawa and Kimura [12] preliminary identified FCCP 1049 from an early Pliocene sediment in Hokkaido, Japan as *Balaenula* sp. based on showing a low triangle shaped occipital with a depression at the center, an acute angle of the nuchal crest against the plain in posterior view, and a rounded exoccipital placing below to the ventral margin of the occipital condyle. The study mentioned that these features can be seen only on *Balaenula balaenopsis*. Bisconti [11] mentioned that FCCP 1049 is different from *Balaenula* based on an extended temporal fossa (in page 47). One of the most recent phylogeny works found that FCCP 1049 does not form a clade with *Balaenula astensis* [2]. To expand diversity and morphological information for understanding the early balaenid evolution, we update identification of FCCP 1049 and its geological age, and we re-describe in this study.

2. Material and methods

Morphological terminology follows Mead and Fordyce [13].

2.1. Institutional abbreviations

FCCP, Fukagawa City Cultural Properties, Hokkaido, Japan. IMNH, Icelandic Museum of Natural History. MLP, Museo de La Plata, La Plata, Argentina. USNM, National Museum of Natural History, Smithsonian Institution, Washington D.C.

3. Systematic paleontology

Cetacea Brisson, 1762

Neoceti Fordyce and de Muizon, 2001

Mysticeti Gray, 1864

Chaemysticeti Mitchell, 1989

Balaenidae Gray, 1825

“*Balaenula*” van Beneden, 1872

Type species. Balaenula balaenopsis

“*Balaenula*” *dosanko* sp. nov

LSID

Holotype. FCCP 1049, including the premaxilla, maxilla, frontal, parietal, squamosal, exoccipital, supraoccipital, periotics, tympanic bullae, right mandible, two thoracic vertebrae, a caudal vertebra, presternum and five ribs. *FCCP 1049 was previously registered as HUES (Hokkaido University of Education Sapporo Campus) 100003.*

Furusawa and Kimura [12] preliminary identified as *Balaenula* sp. In this study, FCCP 1049 is still preliminary belonged to the genus *Balaenula*, because that its phylogenetic position is among an unsolved polytomy. See more in discussion.

Locality and horizon. FCCP 1049 was found at a river bed of Tadoshi River in Fukagawa

City, Hokkaido, Japan by J. Takahashi, N. Mita and others in 22 September 1978 and dug up in 1979 and 1980 by over 100 of people [19]: Latitude 43°47'36.31"N, longitude 142°5'10.00"E (Figure 1).

FCCP 1049 was found from lower part of the Chippubetsu Formation, Fukagawa Group, especially lower to so-called T1 tuff layer [20]. Regarding the study, above T1 tuff layer, there is T2 tuff layer, which is correlated to S1 tuff of the Takikawa Formation. The age of S1 tuff is 4.1 ± 0.6 Ma [21]. T1 tuff possibly distributes above Ops tuff of upper part of the Horokaoshirarika Formation [20]. The age of Ops tuff is 4.5 ± 0.7 Ma. Thus, the age of FCCP 1049 can be taken as 3.5 to 5.2 Ma (Zanclean, early Pliocene) with wider ranges. Very near from the locality (about 10 to 20 km away), *Numataphocoena yamashitai* and *Herpetocetinae* gen. et sp. indet. and some Mysticeti indet. materials from the early Pliocene (about 4.5 to 3.5 Ma) possibly be cetaceans in the same age [22–26]. From the same area, there are some late Miocene mysticeti reports: type specimen of *Miobalaenoptera numataensis* (6.5 to 6.8 Ma) and a referred specimen *Herpetocetus* sp. (7.7 to 6.8 Ma), but are not simultaneous records of FCCP 1049 [27,28].

Etymology. Dosanko means people and things born in Hokkaido, northern Japan, originated from native horse of Hokkaido.

Diagnosis. Among the Balaenidae, "*Balaenula*" *dosanko* uniquely has slender zygomatic process, and deep promontorial groove of the pars cochlearis of the periotic (Character 151). "*B.*" *dosanko* can be differentiated from other balaenids except *Morenocetus parvus* by having a posteriorly oriented postorbital process in dorsal view (Character 38). "*B.*" *dosanko* can be differentiated from balaenids except the *Balaena* and *Eubalaena* by having a weakly and laterally projected lateral margin of the nuchal crest.

Comparison with more basal balaenids (*Morenocetus parvus* and *Peripolocetus vexillifer*), "*B.*" *dosanko* can be differentiated by having a more or less the same length of the anterior process and pars cochlearis of the periotic (Character 139), hypertrophied and blade-like lateral tuberosity (Character 144), posterior process of the periotic orienting a right angle to the axis of the anterior process in ventral view (Character 170), laterally reduced involucral ridge in dorsal view (Character 181), dorsolateral surface of involucrum forming a continuous rim (Character 190), and flat anteromedial portion of ventral surface of tympanic bulla (Character 195). Comparison with crown balaenids (*Balaena* spp. and *Eubalaena* spp.), "*B.*" *dosanko* can be differentiated by having a thickened and flat lateral surface of the orbital rim of the supraorbital process (Character 40), pyramidal process (Character 141), laterally exposed and distinct compound posterior process from the lateral skull wall (Character 172), no crest on the parieto-squamosal suture (Character 93), no hypertrophied suprameatal fossa (Character 162) and no transverse creases on the dorsal surface of the involucrum (Character 191).

4. Description

4.1. Ontogeny

FCCP 1049 possibly be subadult. A caudal vertebra has fused epiphyses. Thoracic vertebrae are not fused with epiphyses. Isolated vertebral epiphyses are preserved. On the skull, the exoccipital/supraoccipital suture is fused, which suggest that FCCP 1049 is older than postnatal [29]. The frontal/parietal, parietal/squamosal, squamosal/exoccipital sutures are fused but visible.

4.2. Skull

General features of the skull. A preserved left side of the skull has a slender supraorbital process with weakly projected pre and postorbital processes, slender zygomatic process and laterally weakly protruded nuchal crest.

Premaxilla. A supposed ascending process is preserved. The ascending process is wide and flat (about 87 mm wide, Figs. 2 and 6). Its posterior still has matrix and does not show the posterior border.

Maxilla. The orbital plate covers anterior border of the supraorbital process. The plate has a transverse ridge on the dorsal surface.

Frontal. The supraorbital process is slender and tilts down laterally. The mid part of the supraorbital process is anteroposteriorly shorter, and the lateral part is anteroposteriorly

longer, because the posterior border of the supraorbital process is anteriorly excavated at the middle (Fig. 2). There is a weak transverse ridge on the dorsal surface of the supraorbital process. The lateral margin of the process in dorsal view is more or less straight. In lateral view (Fig. 5), an anteroposteriorly thin postorbital process projects posteroventrally, and is rounded in dorsal view. The preorbital process is more robust than the post-orbital process. Between the preorbital and postorbital processes form a dorsally excavated orbital region. At the medial end of the frontal contacts with the parietal posteriorly at the anterior portion of the temporal fossa, and with supraoccipital dorsally as a part of the nuchal crest. The frontal forms a part of the vertex, and is partially covered by the ascending process of the maxilla. Medial to the ascending process, the frontal continues medial, which suggests that the left and right frontals connect each other at the midline. The frontal ridge might be absent, not like *Morenocetus parvus* [2]. In ventral view, low preorbital and postorbital ridges run from medial to lateral, and its lateral part curve posteriorly. The parietal locates posterior to the frontal and anterior to the squamosal. The parietal forms the anterior part of the temporal fossa (Fig. 5), which is flattened. The parietal/squamosal suture is visible. The alisphenoid is not visible. **Squamosal** The squamosal has a slender zygomatic process (Fig. 2). The anterior part of the zygomatic process is anteroposteriorly thin, and forms a flat anterior surface for the postorbital process. On the dorsal surface of the zygomatic process, a strong supramastoid crest runs to posteromedially and reaches posterior part of the nuchal crest. The supramastoid crest has a huge anteroposteriorly thin rounded squamosal prominence, dorsal to the postglenoid process and external auditory meatus. The length of the crest is about 170 mm and 65 mm high. On the lateral surface of the base of the zygomatic process shows two square shallow fossae for the sternocephalicus.

In ventral view (Fig. 3), the squamosal has a very faint falciform process, which could be identified using the periotic (Fig. 7). Anterior to the falciform process, there is an anteroposteriorly long shallow groove, which might be a part of foramen ovale. A shallow fossa for the periotic locates medial to the falciform process.

Posterior to the falciform process and lateral to the fossa for the periotic, a large external auditory meatus (23.5 mm long, 26.0+ deep at the lateral end of the meatus) runs transversely. Posterior to the external auditory meatus, there is a huge laterally wider fossa for the posterior process of the periotic (29 mm long, 32 mm high at the lateral end of the fossa). The fossa shows the border between the squamosal and exoccipital. A broken postglenoid process is visible even its anterior and ventral parts are broken away in posterior view (Fig. 4 (a), (b)).

Exoccipital In dorsal and posterior view (Figs. 2 and 4 (a), (b)), the exoccipital forms a rounded ventrolateral part of the occipital shield. In ventral view, the exoccipital is mediolaterally long plate (about 27 mm long), but the medial end is strongly worn. The lateral part of the exoccipital forms a posterior part of the fossa for the posterior process. The occipital condyle is flat and only weakly project posteriorly.

Supraoccipital The supraoccipital is a long triangle (Fig. 2). There is a shallow condyloid fossa dorsal to the occipital condyle. The foramen magnum preserves dorsal side and its dorsal margin becomes thicker.

4.3. Periotic

The periotics (Figs. 7-10 and Table 2) have robust anterior process, small globular pars cochlearis, and large compound posterior process of tympanoperiotic.

The anterior process is short but wide with a prominent lateral tuberosity. The size and shape of the lateral tuberosities and also the posterior processes are slightly different on the right and left sides of FCCP 1049 (Table 2). Anterior to the lateral tuberosity, there is a shallow notch, which might be the anterioexternal sulcus. Medial to the lateral tuberosity, the anterior pedicle of the tympanic bulla is anteroposteriorly long rectangular, and places on the ventral surface of the anterior process. Between the anterior pedicle and lateral tuberosity, there is a large and weakly depressed plane for the sigmoid process of the tympanic bulla (Fig. 8). A shallow wide malleolar fossa is located slightly posterior level of the lateral tuberosity. In ventral view, the medial part of the periotic has two large processes: the pyramidal process anteriorly and the dorsal tuberosity posteriorly. Between these processes,

an anteroposteriorly long and strongly curved dorsal rim of the suprameatal fossa [30]. Ventral to the rim, the suprameatal fossa is large and shallow.

An anteroposteriorly long globular pars cochlearis covers the body of the periotic anteriorly as a thin bone and it forms the anterior incisure. In the anterior incisure, there is a huge dorsoventrally long elliptical hiatus fallopii (4.7 mm high, 3.2 mm wide) opening anteriorly. The medial surface of the pars cochlearis has a deep, anteroposteriorly long and weakly curved promontorial groove. Just posterior to the hiatus fallopii, a large internal acoustic meatus (9.1 mm long, 5.2 mm high) opens medially and contains the proximal opening of the facial canal and dorsal vestibular area. The proximal opening of the facial canal opens (5.0 mm high, 4.3 mm long). Just posterior to the facial canal, and clearly separated from it by a low transverse crest, there is a circular dorsal vestibular area (5.2 mm high, 5.9 mm long). In the dorsal vestibular area, there is a small foramen singulare (1.0 mm diameter) anterodorsally and two depressed areas, which might be the spiral cribriform tract and area cribrosa media posteroventrally. The spiral cribriform tract and area cribrosa media are connected and form a weakly curved depression. Posterior to these openings, there are two foramina: a larger vestibular aqueduct (4.0 mm high, 4.8 mm long), and a smaller aperture for the cochlear aqueduct (2.5 mm high, 1.2 mm long). The posterior end of the pars cochlearis has a large elliptical fenestra rotunda (4.4 mm long, 8.1 mm wide), and the anterior border of the fenestra rotunda runs to the medial end of the pars cochlearis and projects as an anteroposteriorly short plate. The posterior end of the pars cochlearis is formed by the caudal tympanic process, which has transversely long sharp edge.

A large fenestra ovalis (6.0 mm long, 4.3 mm wide) is located posterior to the malleolar fossa, lateral to the pars cochlearis (Fig. 10, (i)). Posterior to the fenestra ovalis, there is a deep and long stapedial muscle fossa (15.5 mm long, 4.5 mm wide). Medial to the fenestra ovalis, a long fossa for the anterior part of the facial sulcus. The facial sulcus runs posteriorly, and forms a deep groove on the posterior surface of the compound posterior process of the tympanoperiotic. Lateral to the facial sulcus, an anteroposteriorly long elliptical posterior pedicle of the tympanic bulla shows broken surface. The posterior process becomes dorsoventrally high at the middle of the process, and the lateral end is dorsoventrally thin.

4.4. Tympanic bullae

The tympanic bullae (Figs. 7, 8, 11, 12 and Table 2) are small and wide with wide anterior portion in ventral view.

The anterior robe of the tympanic bulla is swollen laterally. On the ventral to lateral surface of the anterior robe has a blunt ridge, which is a part of the main ridge running to the medial surface. Posterior to the anterior robe, a large triangular sigmoid process projects laterally. Its dorsal most point bends posteriorly (Fig. 12 (a)). The sigmoid process has thick margins, about 8.0 mm (anteroposterior length). Anterior to the sigmoid process, a deep lateral furrow runs transversely and separates the sigmoid process and anterior robe. Anterior to the sigmoid process, the rim of the outer lip becomes thicker, which is the malleolar ridge. The malleolar ridge contains structures such as the sulcus for the chorda tympani and fossa for the malleus dorsally. An anteroposteriorly long sulcus for the chorda tympani locates anterior to a long fossa for the malleus. The lateral surface of the malleolar ridge has several oblique striae. The involucrem is posteriorly wider (Fig. 12, (g)). Its lateral surface has several transverse grooves. A broken base of the posterior pedicle locates at the posterior end of the involucrem on the lateral surface. The broken section of the posterior pedicle is anteroposteriorly long elliptical (16.6 mm long, 9.0 mm wide). Medial to the posterior pedicle, there is a smooth conical process (Fig. 12, (g)). On the medial surface of the tympanic bulla, there are stronger main ridge and weaker involucrem ridge running anteroposteriorly. Between them, there is a median furrow, which is deep anteriorly but weak posteriorly. The main ridge runs from the anterior lobe to the posterior end of the tympanic bulla, forming the anterior most point and also the medial margin of the tympanic bulla.

4.5. Mandible

An incomplete right mandible is weakly laterally bowed, especially anterior part (Figure 13, Table 1). The most anterior part has an anteroposteriorly long mental foramen, which also

opens anteriorly. The anterior part of the mental foramen is shallower and wider. Medially, there is a flat area for mandibular symphysis, which is restricted by a wide symphyseal groove running anteroposteriorly. The symphyseal groove is anteriorly wider and shallower, and its posterior end runs along the medial surface of the mandible. The posterior part preserves a broken coronoid process. The mandibular foramen (about 50 mm high and 40 mm wide at the broken posterior margin) locates dorsally in medial view.

4.6. Postcranial skeleton

Vertebra. Two thoracic vertebrae (Fig. 14 (a) and (b)) show dorsally wider triangular body with a ventral keel, large neural canal and long transverse process. A fossa for the rib locates ventral to the transverse process. The vertebral epiphyses are all removed. A caudal vertebra (Fig. 14 (c)) shows the body with fused epiphyses. Supposed anterior articular surface is slightly larger than posterior one (anterior surface: 86 mm wide and 93 mm high, posterior surface: 80 mm wide and 90 mm high, length of the body is 71 mm). Seven isolated epiphysis (Fig. 14 (d) to (j)) are dorsally wider triangular to laterally wider elliptical. The smallest epiphysis number D is 90 mm wide and 81 mm high. The largest epiphysis number J is 131 mm wide and 112 mm high.

Presternum. Preserved part of the presternum shows that symmetrical anterior part is broken and dorsoventral surfaces are worn (Fig. 14, (k), Table 3). A side is convex, which can be identified as the ventral side (Fig. 14, (k)). On the lateral margin, there are a couple of thicker part, which might be a surface for the ribs.

Rib. Five ribs are preserved (Fig. 15). The rib number (b) in Figure 15 is wide, which might be the most anterior rib. The longest one, number E is about 560 mm long and 65 mm wide, and its restoration using clay might overestimate its length.

5. Discussion

5.1. Phylogenetic analysis

The phylogenetic position of “*Balaenula*” *dosanko* sp. nov. (FCCP 1049) was analyzed using the matrix of Bueno et al. [2], which was derived from the matrix of Marx and Fordyce [31]. This study modified codings of FCCP 1049 with direct examination (Supplementary 1 to 4) and contains 257 morphological characters and 43 taxa. Percentages of coded data of FCCP 1049 are 46 % (includes soft tissue characters), 48 % (excludes soft tissue) and 89% for the ear bones.

The matrix was managed using Mesquite 2.75 [32]. Analysis was performed with TNT version 1.5 [33]. All of the characters were treated as unweighted and unordered with backbone constraint of extant taxa, based on a topology of the molecular tree [34]. The analysis used New Technology Search with recover minimum length trees = 1000 times.

The phylogenetic analysis shows two shortest trees of 947 steps each. The strict consensus trees (Fig. 16 and Supplementary 5) are different from that of the equally weighted analysis of Bueno et al. [2] in the most basal balaenid is *Moreno* *is parvus* but *Peripolocetus vexillifer*, and “*B.*” *dosanko* + *Balaenula astensis* + *Balaenella brachyrhynchus* + crown clade are appeared as an unsolved polytomy. Indeed, the branch lengths of the previous phylogenetic hypothesis around them were relatively low [2].

5.2. Comparison with *Balaenula* and *Balaenella*

“*Balaenula*” *dosanko* is identical from the all named balaenids, but hard to determine the genus belonged. Because the results of this phylogenetic analysis show “*Balaena*” *dosanko* + *Balaenula astensis* + *Balaenella brachyrhynchus* + crown clade as an unsolved polytomy. Here, we compare “*B.*” *dosanko* with the *Balaenula* then *Balaenella* (Table 4).

Several diagnoses of the genus *Balaenula* was reported by Bisconti [7]. A diagnosis for the genus, “having a not protruding nasal crest” is not seen on FCCP 1049. FCCP 1049 shows a lateral expansion at the posterior part of the supraoccipital crest. One comparable diagnosis of the genus, such as having a low and rounded supramastoid crest (the lateral squamosal crest in the study) is difficult to consider. Because some *Balaenula* species such as *Eubalaena belgica* also shows low and rounded supramastoid crest (see later). Thus, not only the results of phylogenetic analyses, but also the previously mentioned diagnoses for the *Balaenula* support that “*B.*” *dosanko* might not belong to the *Balaenula*. However,

here we present “*Balaenula*” *dosanko* in the genus, preliminary.

The type species of the genus *Balaenula* is doubtful to recognize as an individual [2]. Indeed, the long and triangular anterior process of the periotic on the type specimen is not likely belongs to the Balaenidae, but Balaenopteroidea. Based on illustrations of previous studies [7,35], the periotic seems having some balaenopteriid diagnoses such as a narrow and triangular anterior process [36,37], but lack other diagnoses for the family such as a well defined fossa for the malleus [37] and transversely elongated pars cochlearis [38]. Thus, it might be a stem balaenopteroid. Its overall shape in ventral view is similar to that of *Tiphyocetus temblorensis* of Kellogg [39] by having a slender anterior process and globular pars cochlearis, but morphologies of the internal acoustic meatus and lateral tuberosities are different.

Here, comparisons on “*B.*” *dosanko* with closely related genus *Balaenula* and *Balaenella* (see Table 4). Comparison with the genus *Balaena*, “*B.*” *dosanko* shows weaker pre and postorbital processes in lateral view, and weaker orbitotemporal crest only on the medial part of the supraorbital process. Comparison with *Balaenula balaenopsis*, “*B.*” *dosanko* shows more or less straight lateral margin of the supraorbital process, and much weaker supramastoid crest. Comparison with *Balaenula astensis*, FCCP 1049 shows a more slender supraorbital process, no tubercle at the junction of the parieto-squamosal suture and supraoccipital (Character 82), lateral projection of the nuchal crest, and more posteriorly located posterior supraoccipital fossa.

Comparison with *Balaenella brachyrhynchus*, “*B.*” *dosanko* shows a more strongly curved posterior margin of the supraorbital process in dorsal view, much weaker supramastoid crest, more rounded postorbital process in dorsal view, ventrally almost closed frontal groove by the pre and postorbital ridges, much smaller lateral process of the periotic, thinner medial margin of the fenestra rotunda, much weaker outer posterior prominence of the tympanic bulla, much stronger inner posterior prominence, and more rounded anteromedial angle of the tympanic bulla in ventral and dorsal views. Of note, *Balaenella brachyrhynchus* does not preserve the anterior tip of the zygomatic process. Some *Balaenula* sp. specimens from the late Miocene to early Pliocene, and Late Pliocene of California have mentioned, but not been photographed and illustrated [40].

5.3. The supramastoid crest

The supramastoid crest of balaenids are generally developed. But, morphological change of the supramastoid crest can not be thought simple, like developing from incipient to large through balaenid evolution. The oldest known balaenid *Morenocetus parvus* shows small supramastoid crest on Holotype, but just a ridge along the dorsal surface of the zygomatic process on referred specimen (MPL 5-15) [2].

The modern balaenids show varied condition of the supramastoid crest. *Balaena mysticetus* shows both small (a juvenile individual described by Nishiwaki and Kasuya (1970)) and well developed (USNM 257513, see figure 5 of Bisconti [7]) conditions, as Field et al. [42] mentioned. *Eubalaena australis* shows small ones (a young individual in Best [43] and USNM 26712, see figure 4 of Bisconti [7]). Possibly the development of the supramastoid crest depends on growing. Some fossil species belonging to the genus *Balaena* and *Eubalaena* such as *Eubalaena elgica* and *E. ianatrix* also have small ridges [3,44], but *B. montalionis* and *Eubalaena* sp. (IMNH 9598) have large ridges [11].

As mentioned above, having a low and rounded supramastoid crest was considered as a diagnosis of the genus *Balaenula*. However, early Pliocene balaenids (“*B.*” *dosanko*, *Balaenula* spp. and *Balaenella brachyrhynchus*) have relatively large supramastoid crests among the Balaenidae. Among them, the conditions of the supramastoid crest are varied as compared above (see also Table 4), especially *Balaenula balaenopsis* has larger and dorsally strongly projected supramastoid crest than those of “*B.*” *dosanko*, *Balaenula astensis* and *Balaenella brachyrhynchus*.

The rounded and low supramastoid crest emerged among the Balaenidae already during the early Miocene as incipient condition, then enlarged by the early Pliocene, and some kept the condition or others got the crest smaller secondary. In short, the morphological change of the supramastoid crest is not simple. The supramastoid crest is attached to the temporal fascia [45]. Thus, these conditions of the supramastoid crest might be related to size

and/or orientation and the temporal muscle, but is also related to many factors (width of the supraorbital process, degree of telescoping, orientation of the skull, arching of the rostrum, width, height and length of the temporal fossa).

6. Conclusion

“*Balaenula*” *dosanko* sp. nov. (FCCP 1049) represents an archaic balaenid from the early Pliocene, Zanclean (3.5 to 5.2 Ma) lower part of the Chippubetsu Formation, Hokkaido, northern Japan. The fossil balaenid is distinguishable from another balaenids by having a slender zygomatic process, deep promontorial groove of the pars cochlearis of the periotic, and a weak orbitotemporal crest only on the medial part of the supraorbital process. “*B.*” *dosanko* does not show previously introduced diagnoses for the genus *Balaenula*. The result of phylogenetic analysis places “*B.*” *dosanko* among an unsolved polytomy of *Balaenula astensis* + *Balaenella brachyrhynchus* + crown clade. “*B.*” *dosanko* is different from these closely related taxa in many points. Thus, “*B.*” *dosanko* is preliminarily belonged to the genus “*Balaenula*”, in this study. The genus *Balaenula* had been treated as “a taxonomical basket, where all the small-sized balaenids were put in. In addition, the type species of the genus is doubtful to recognize as an individual. It might be better to wait taxonomical works by adding more related species in the future.

Data accessibility. The data for the phylogenetic analysis is available as electronic supplementaries.

Authors’ contributions. Y. T. carried out the phylogenetic analysis and write the paper. H. F. and M. K. revised the manuscript.

Competing interests. There are no competing interests.

Funding. There are no fundings.

Acknowledgements. We thank to J. Takahashi, N. Mita and others to discover, and 100 
[revised manuscript text omitted]

Figure 2. Skull of FCCP 1049, “*Balaenula*” *dosanko* in dorsal view. (a) photo; (b) line art.

Figure 3. Skull of FCCP 1049, “*Balaenula*” *dosanko* in ventral view. (a) photo; (b) line art.

Figure 4. Skull of FCCP 1049, “*Balaenula*” *dosanko* in anterior and posterior views. (a) photo; (b) line art; (c) photo; (d) line art.

Figure 5. Skull of FCCP 1049, “*Balaenula*” *dosanko* in lateral view. (a) photo without the supraorbital process in left lateral view; (b) photo with the supraorbital process in left lateral view; (c) photo in right lateral view; (d) line art of (b).

Figure 6. Skull, vertex of FCCP 1049, “*Balaenula*” *dosanko* in dorsal view. (a) photo; (b) line art.

Figure 7. Skull and left ear bones of FCCP 1049, “*Balaenula*” *dosanko* in ventral view. (a) with periotic and without tympanic bulla; (b) with periotic and tympanic bulla.

Figure 8. Right periotic and tympanic bulla connections of FCCP 1049, “*Balaenula*” *dosanko* in ventrolateral view.

Figure 9. Periotics of FCCP 1049, “*Balaenula*” *dosanko*; (a)-(f) left periotic; (g)-(i) right periotic; (a) and (h) ventral view; (b) and (g) medial view; (c) dorsal view; (d) lateral view; (e) posterior view; (f) anterior view; (i) posteroventral view.

Figure 10. Key features of periotics of FCCP 1049, “*Balaenula*” *dosanko*; (a)-(f) left periotic; (g)-(i) right periotic; (a) and (h) ventral view; (b) and (g) medial view; (c) dorsal view; (d) lateral view; (e) posterior view; (f) anterior view; (i) posteroventral view.

Figure 11. Tympanic bullae of FCCP 1049, “*Balaenula*” *dosanko*; (a)-(f) right tympanic

bulla; (a) lateral view; (b) ventral view; (c) anterior view; (d) medial view; (e) posterior
view; (f) dorsal view; (g) left tympanic bulla in lateral view.

**Figure 12.** Key features of tympanic bullae of FCCP 1049; (a)-(f) right tympanic bulla; (a)
lateral view; (b) ventral view; (c) anterior view; (d) medial view; (e) posterior view; (f)
dorsal view; (g) left tympanic bulla in lateral view.

**Figure 13.** Right mandible of FCCP 1049, "*Balaenula*" *dosanko*; (a) lateral view; (b) dorsal
view; (c) medial view; (d) to (f) cross sections; (g) medial view of posterior end.

**Figure 14.** Postcranial skeleton of FCCP 1049, "*Balaenula*" *dosanko*; (a) and (b) thoracic
vertebrae; (c) caudal vertebra; (d) to (j) isolated vertebral epiphyses; (k) presternum in
ventral view.

**Figure 15.** Ribs of FCCP 1049, "*Balaenula*" *dosanko* (a) to (c) right ribs; (d) and (f) left
ribs.

**Figure 16.** Strict consensus tree of two most parsimonious trees to show the phylogenetic
position of "*Balaenula*" *dosanko* sp. nov. The most parsimonious trees were 974 steps. The
clades Aetiocetidae, Eomysticetidae, Balaenidae, Cetotheriidae and a clade comprising
*Isanacetus*, *Parietobalaena* and related taxa are collapsed. (See Supplemental material file 5
for cladogram with all taxa shown.) The numbers on the each branch are branch lengths. The
numbers next to the branch lengths show the percentages of the shortest trees supporting
each node. 100% supported nodes are omitted the percentages.

**Table 1.** Measurements in cm of "*Balaenula*" *dosanko* (FCCP 1049) skull and mandible
following Buono *et al.* [2], and Tanaka and Taruno [48]. For skull and mandible, distances
are either horizontal or vertical, unless identified as point to point. + shows an incomplete
measurement, because of erosion. * shows measurements taken from only one side.

**Table 2.** Measurements in mm of "*Balaenula*" *dosanko* (FCCP 1049) periotic and tympanic
bulla. Dimensions follow [49]. Distances are either horizontal or vertical, unless identified as
point to point. + shows an incomplete measurement, because of erosion. Measurements of
*Balaenulla balaenopsis* CtM 858a was taken from table 7 of Bisconti [7].

**Table 3.** Measurements in mm of "*Balaenula*" *dosanko* (FCCP 1049) vertebrae and
presternum. Dimensions follow Tanaka and Taruno (submitted). Measurements are rounded
to the nearest 0.5 cm. Distances are either horizontal or vertical, unless identified as point to
point. + shows an incomplete measurement, because of erosion.

**Table 4.** Comparison between "*Balaenula*" *dosanko* (FCCP 1049) and related fossil
balaenids using FCCP 1049 specimen and previous publications [4,35].

**Supplementary 1.** Datamatrix in nex format.

**Supplementary 2.** Datamatrix in tnt format.

**Supplementary 3.** Character list.

**Supplementary 4.** Modified codings from Buono et al (2017).

**Supplementary 5.** Full figure of the analysis

**Supplementary 6.** Tree file

Figure 1. Maps showing the locality of FCCP 1049, "Balaenula" dosanko The base maps A and B are modified from Tanaka and Kohno [46], and Wada and Akiyama [47].

183x90mm (300 x 300 DPI)

Figure 2. Skull of FCCP 1049, "Balaenula" dosanko in dorsal view. (a) photo; (b) line art.

189x237mm (300 x 300 DPI)

Figure 3. Skull of FCCP 1049, "Balaenula" dosanko in ventral view. (a) photo; (b) line art.

187x241mm (300 x 300 DPI)

Figure 4. Skull of FCCP 1049, "Balaenula" dosanko in anterior and posterior views. (a) photo; (b) line art; (c) photo; (d) line art.

188x163mm (300 x 300 DPI)

Figure 5. Skull of FCCP 1049, "Balaenula" dosanko in lateral view. (a) photo without the supraorbital process in left lateral view; (b) photo with the supraorbital process in left lateral view; (c) photo in right lateral view; (d) line art of (b).

189x161mm (300 x 300 DPI)

Figure 6. Skull, vertex of FCCP 1049, "Balaenula" dosanko in dorsal view. (a) photo; (b) line art.

182x60mm (300 x 300 DPI)

Figure 7. Skull and left ear bones of FCCP 1049, "Balaenula" dosanko in ventral view (a) with periotic and without tympanic bulla; (b) with periotic and tympanic bulla

94x88mm (300 x 300 DPI)

Figure 8. Right periotic and tympanic bulla connections of FCCP 1049, "Balaenula" dosanko in ventrolateral view.

89x115mm (300 x 300 DPI)

45 Figure 9. Periotics of FCCP 1049, "Balaenula" dosanko; (a)-(f) left periotic; (g)-(i) right periotic; (a) and (h)
46 ventral view; (b) and (g) medial view; (c) dorsal view; (d) lateral view; (e) posterior view; (f) anterior view;
47 (i) posteroventral view.

185x236mm (300 x 300 DPI)

Figure 10. Key features of periotics of FCCP 1049, "Balaenula" dosanko; (a)-(f) left periotic; (g)-(i) right periotic; (g) and (h) ventral view; (b) and (g) medial view; (c) dorsal view; (d) lateral view; (e) posterior view; (f) anterior view; (i) posteroventral view.

187x236mm (300 x 300 DPI)

Figure 11. Tympanic bullae of FCCP 1049, "Balaenula" dosanko; (a)-(f) right tympanic bulla; (a) lateral view; (b) ventral view; (c) anterior view; (d) medial view; (e) posterior view; (f) dorsal view; (g) left tympanic bulla in lateral view.

187x235mm (300 x 300 DPI)

Figure 12. Key features of tympanic bullae of FCCP 1049; (a)-(f) right tympanic bulla; (a) lateral view; (b) ventral view; (c) anterior view; (d) medial view; (e) posterior view; (f) dorsal view; (g) left tympanic bulla in lateral view.

187x236mm (300 x 300 DPI)

Figure 13. Right mandible of FCCP 1049, "Balaenula" dosanko; (a) lateral view; (b) dorsal view; (c) medial view; (d) to (f) cross sections; (g) medial view of posterior end.

184x99mm (300 x 300 DPI)

Figure 14. Postcranial skeleton of FCCP 1049, "Balaenula" dosanko; (a) and (b) thoracic vertebrae; (c) caudal vertebra; (d) to (j) isolated vertebral epiphyses; (k) presternum in ventral view.

187x61mm (300 x 300 DPI)

Figure 15. Ribs of FCCP 1049, "Balaenula" dosanko (a) to (c) right ribs; (d) and (e) left ribs.

63x115mm (300 x 300 DPI)

Figure 16. Strict consensus tree of two most parsimonious trees to show the phylogenetic position of "Balaenula" dosanko sp. nov. The most parsimonious trees were 974 steps. The clades Aetiocetidae, Eomysticetidae, Balaenidae, Cetotheriidae and a clade comprising Isanacetus, Parietobalaena and related taxa are collapsed. (See Supplemental material file 5 for cladogram with all taxa shown.) The numbers on the each branch are branch lengths. The numbers next to the branch lengths show the percentages of the shortest trees supporting each node. 100% supported nodes are omitted the percentages.

174x54mm (300 x 300 DPI)

Skull	
Bizygomatic width	59.0*
Width of the skull at the level of the exoccipitals	34.0*
Supraoccipital width anterior to foramen magnum	32.2*
Supraoccipital width at mid-length	18.0*
Supraoccipital width at 10 cm to the anterior margin	12.0*
Supraoccipital length	38.8
Transverse diameter of left occipital condyle	7.8
Dorsoventral diameter of left occipital condyle	9.4+
Transverse diameter of foramen magnum	7.2
Distance between lateral border of left occipital condyle	22.0
Width of the occipital condyles plus foramen magnum	18.8 plus
Anteroposterior diameter of the supraorbital process	16.0
Transverse diameter of the supraorbital process	48.2
Anteroposterior diameter of the supraorbital process	10.2
Width of the optic canal in its medial portion	0.7
Width of the optic canal in its lateral portion	11.2
Length of the optic canal	40.0+
Orbital length (between preorbital and postorbital foramina)	13.0
Anteroposterior diameter of the temporal fossa	27.5
Transverse diameter of the temporal fossa	39.5
Mandible	
Length of mandible, as preserved in straight line	180.0
Height of mandible, from coronoid process to ventral margin	15.5+
Maximum preserved height of mandible	15.5+
Maximum preserved width of mandible	74.0
Height of body at just anterior to coronoid process	15.3
Width of body at just anterior to coronoid process	74.0

	left	right	Balaenul la balaeno
Periotic			
Length of anterior process of periotic	34.7	36.0	32.0
Width of anterior process of periotic	46.4	60.0	43.5
Length of anterior process, from anteroventral angle to anterior incisure	24.3	32.7 -	
Maximum transverse width of the body of periotic	46.4	60.0 -	
Length of pars cochlearis	30.2	31.4	54.0
Width of pars cochlearis	16.8	17.7	30.0
Dorsoventral diameter of internal acoustic meatus			
	9.3	9.3	4.3
Length of internal acoustic meatus	13.7	13.9	6.8
Greatest length of aperture for the cochlear aqueduct	1.2 -	-	
Greatest length of aperture for the vestibular aqueduct	4.8	5.0 -	
Dorsoventral diameter of fenestra rotunda			
	4.1 -		3.4
Width of fenestra rotunda	7.5 -		4.3
Length of fenestra ovalis	6.0 -		3.7
Width of fenestra ovalis	4.3 -		5.4
Greatest length of malleolar fossa	7.0	5.6 -	
Greatest length of ventral opening of the facial canal tuberosity	-	26.4 -	
	42.7	54.4 -	
Length of the posterior process of the periotic	139.6	114.6 -	
Bulla			
Greatest length, in lateral view	94.7	94.4 -	
Greatest length of tympanic cavity	-	72.8 -	
Greatest width, in lateral view	-	74.2 -	
Greatest height, from tip of sigmoid process to the ventral-most point in posterior view	-	65.2 -	
Height of involucre at the anteriormost point of the posterior pedicle	30.8	34.0 -	
Length of anterior lobe, from lateral furrow to anterior tip of tympanic bulla	-	42.0 -	
Greatest length of posterior pedicle at base	16.6	16.9 -	

Thoracic vertebra A in Fig. 13

maximum preserved length	77.0
maximum preserved height	270.0
maximum preserved width	191.0+
length of neural spine	48.0
height of neural spine	115.0
length of body	52.0
width of body anteriorly	122.0
height of body anteriorly	92.0
width of body posteriorly	144.0
height of body posteriorly	90.0

Thoracic vertebra B in Fig. 13

maximum preserved length	86.0
maximum preserved height	172.0
maximum preserved width	237.0
length of body	49.0
width of body anteriorly	125.0
height of body anteriorly	77.0
width of body posteriorly	132.0
height of body posteriorly	92.0

Sternum

maximum preserved length	118.0
maximum preserved height	18.0
maximum preserved width	83.0

Sheet1

	"Balaenula" dosanko	Balaenula balaenopsis	Balaenula astensis	Balaenella brachyrhynus
pre and postorbital processes	weak	strongly project	strongly project	weak
lateral margin of supraorbital process in supraorbital process	straight	strongly excavated	straight	straight
orbitotemporal crest	weak, only on medial part of supraorbital	strong	strong	weak, only on medial part of supraorbital process
supramastoid crest	weaker	stronger	weaker	weaker
zygomatic process	slender	robust	robust	-
Distinct tubercle at jugular notch	absent	-	absent	present
lateral margin of nuchal crest	laterally projected	-	straight	laterally projected
fossa	absent	-	present	absent
posterior supraoccipital fossa position	just anterior to occipital condyles	-	more anterior	-

Dear Sir/Madam,

Manuscript RSOS-192182 entitled "A new member of fossil balaenid (Mysticeti, Cetacea) from the early Pliocene of Hokkaido, Japan"

Thank you for your email for the acceptance. We have revised following the all comments from the editor and reviewers. All changes are recorded using track change function, and there are no changes in the abstract and contents of the manuscript.

- Ethics statement

We do not use humans or animals' materials.

- Data accessibility

<http://datadryad.org/submit?journalID=RSOS&manu=RSOS-192182>

We upload 6 supporting files.

- Competing interests

None of the authors have competing interests.

- Authors' contributions

YT participated in conceptualization, methodology analysis, investigation, writing of original draft. HF

and MK contributed review and editing.

- Acknowledgements

Please acknowledge anyone who contributed to the study but did not meet the authorship criteria. We added the reviewers this time.

- Funding statement

The research was not funded.

Done.

Reviewer: 1 Toshiyuki Kimura

Comments to the Author(s)

1) There are several typos in the text. Please double check.

Fixed.

2) The authors preliminary diagnosed the specimen (FCCP 1049) as "Balaenula" dosanko, sp. nov. The tentative generic combination is allowed by ICZN. However, in this paper, the authors clearly mention that FCCP 1049 is differed from the other balaenid (e.g., page4 lines 20-25). I think that the authors should explain on why they do not describe FCCP 1049 as new genus.

Following recommendations by reviewers 1 and 2, we provide new genus name.

3) The authors preliminary diagnosed FCCP 1049 as genus "Balaenula". But, they do not adequately explain on the reason for their (preliminary) decision. Even if it is a preliminary decision, I think the authors should justify their decision based on the morphological characters found in the specimen.

Following recommendations by reviewers 1 and 2, we provide new genus name.

4) The authors mentioned that ["Balaenula" dosanko is hard to determine the genus belonged. Because the result of this phylogenetic analysis show clade as an unsolved polytomy.] (page 7. lines 48-51). I think that the taxon is not defined by the topology of the tree itself, but the taxon is defined by the characters (e.g., ICZN Art. 13.1.1: "every new name [of taxon] must be accompanied by a description or definition that states in words characters that are purported to differentiate the taxon). The authors clearly mention that FCCP 1049 is differed from the other balaenid (e.g., page4 lines 20-25). I would like to recommend the authors to describe the specimen as new genus.

Following recommendations by reviewers 1 and 2, we provide new genus name.

Reviewer: 2 Mónica R. Buono

My main concerns are in the following sections of the manuscript:

1) Introduction: this section needs to be improved in order to be more informative for readers. Some references should be added and there are many comments and corrections I have suggested in the pdf file.

We added more information.

2) Diagnosis and description:

-Diagnosis: some characters should be revised considering they are not apomorphic for this taxon

(i.e. morphology of zygomatic process). I suggest that the authors first work in a differential diagnosis in order to better determine if the specimen is a new genus or if it could be maintained inside of *Balaenula*. I think that this would be more important than providing a new species name. There is enough anatomical evidence to differentiate this specimen from *Morenocetus*, *Balaena*, *Eubalaena* and *Balaenella*, considering not only skull but also the tympano-periotic morphology (I suggest the authors should focus on that). Then, continue to present if there is enough evidence to nominate a new species. More comments in the pdf file.

Following recommendations by reviewers 1 and 2, we provide new genus name. Also revised diagnoses for the new taxon.

-Description: I have made many suggestions in the description section, some of them include the revision of some anatomical structures which should be taken into account by the authors.

Fixed.

3) Phylogenetic analysis:

My main concern with this point is that authors perform only one analysis, using a molecular constraint, which shows "*Balaenula*" *dosanko* in a polytomy. Why did the authors not explore other types of phylogenetic analysis? For example, without the molecular constraint (and between equal and implied weights?). Besides, the type species *Balaenula balaenopsis* should be added to the analysis, especially considering the authors are testing the validity of a *Balaenula* species's. Please also check my comments in the supplementary file S4 modified coding.

We changed some codings after reading comments of the reviewer.

4) Discussion.

One of the main problems that I saw in the paper is the taxonomic status of the specimen studied (FCCP 1049). Many sections of the manuscript pointed out that the assignation to genus *Balaenula* is not well supported (for example, there are no characters supporting its diagnosis and also the taxon is recovered in a polytomy with *B. astensis* and *Balaenella*). The re-diagnosis provided by Bisconti (2003) included other characters not discussed by the authors. Besides, the phylogenetic analysis did not include the type species of *Balaenula*. I understand the status of this taxon is conflictive, however, is hard to really test the validity of this species and its assignation to *Balaenula* if the type species is not included in the analysis. Why did the authors decide to maintain this assignation?

We changed some codings after reading comments of the reviewer.

I recommend to the authors to revise the phylogenetic analysis, include *B. balaenopsis* (at least with the materials which are more reliable such as the skull and skeleton) and include deeper comparisons with both taxa: *B. balaenopsis* and *B. astensis*. It should be noted that *Balaenula astensis* morphology should be interpreted with caution because the specimen exhibits many juvenile characters. In my opinion, skull characters are not fully useful to diagnose *Balaenula*; periotic and bulla morphology should be further considered, in particular between "*Balaenula*" *dosanko* and *B. astensis*. Thorough comparisons of the tympano-periotic morphology should be made in order to discuss differences or similarities, which could better support (or not) the assignation of FCCP 1049 to *Balaenula*.

We could not include *B. balaenopsis* because the specimen possibly contains some individuals, as the reviewer previously mentioned in her article.

Finally, I suggest to further expand the discussion of the evolution of supramastoid crest in balaenids (see comments on the pdf).

We expanded this point.

Figures:

Generally, the figures are fine, and of high quality. However, there are spelling mistakes to be corrected (see pdf). The scales in the figures should be consistently placed in all figures in the same place.

Fixed.

Figure 6: it needs revision because there are structures (premaxilla vs maxilla) that are not consistently interpreted as in the description section of the manuscript.

Fixed.

Tables:

Please check the format of the tables (different font and font size are observed). Besides, many words appear to be cut.

Fixed.

Sincerely,

Yoshi Tanaka 嘉

Yoshihiro TANAKA, Ph. D.

Curator of Geohistory

Osaka Museum of Natural History

Nagai Park 1-23, Higashi-Sumiyoshi-ku, Osaka, 546-0034, JAPAN

Tel: +81-(0)6-6697-6222

Email: tanaka@mus-nh.city.osaka.jp